# Disruption of Nrxn1α within excitatory forebrain circuits drives value-based dysfunction

Opeyemi O Alabi[1,2], M Felicia Davatolhagh[1,2], Mara Robinson[1], Michael P Fortunato[1], Luigim Vargas Cifuentes[1,2], Joseph W Kable[3], Marc Vincent Fuccillo[1]*

[1]Department of Neuroscience, Philadelphia, United States; [2]Neuroscience Graduate Group, Perelman School of Medicine, Philadelphia, United States; [3]Department of Psychology, University of Pennsylvania, Philadelphia, United States

**Abstract** Goal-directed behaviors are essential for normal function and significantly impaired in neuropsychiatric disorders. Despite extensive associations between genetic mutations and these disorders, the molecular contributions to goal-directed dysfunction remain unclear. We examined mice with constitutive and brain region-specific mutations in Neurexin1α, a neuropsychiatric disease-associated synaptic molecule, in value-based choice paradigms. We found Neurexin1α knockouts exhibited reduced selection of beneficial outcomes and impaired avoidance of costlier options. Reinforcement modeling suggested that this was driven by deficits in updating and representation of value. Disruption of Neurexin1α within telencephalic excitatory projection neurons, but not thalamic neurons, recapitulated choice abnormalities of global Neurexin1α knockouts. Furthermore, this selective forebrain excitatory knockout of Neurexin1α perturbed value-modulated neural signals within striatum, a central node in feedback-based reinforcement learning. By relating deficits in value-based decision-making to region-specific Nrxn1α disruption and changes in value-modulated neural activity, we reveal potential neural substrates for the pathophysiology of neuropsychiatric disease-associated cognitive dysfunction.

*For correspondence:
fuccillo@pennmedicine.upenn.edu

Competing interests: The authors declare that no competing interests exist.

## Introduction

Goal-directed behaviors are a critical aspect of animal fitness. Their implementation engages widespread neural circuits, including cortico-striatal-thalamic loops and midbrain dopaminergic populations. Cortical regions including orbital frontal (OFC), medial prefrontal (mPFC), and anterior cingulate (ACC) represent aspects of reward value and history (*Bari et al., 2019*; *Bartra et al., 2013*; *Euston et al., 2012*; *Noonan et al., 2011*; *Padoa-Schioppa and Conen, 2017*; *Rushworth et al., 2011*; *Rushworth et al., 2012*). Primary sensory cortices and midline thalamic nuclei represent reward-associated environmental signals (*Parker et al., 2019*; *Znamenskiy and Zador, 2013*) while motor thalamic nuclei ensure smooth performance of actions (*Díaz-Hernández et al., 2018*). Furthermore, flexible adaptation of value signals is supported by error-monitoring signals within ACC and basolateral amygdala, as well as reward prediction errors encoded by striatal-targeting midbrain dopaminergic neurons (*McGuire et al., 2014*; *Schultz et al., 1997*; *Ullsperger et al., 2014*; *Yacubian et al., 2006*). The dorsal striatum via integration of these diverse projections can simultaneously mediate action selection, motor performance, and reinforcement learning (*Balleine et al., 2007*; *Cox and Witten, 2019*; *Lee et al., 2015*; *Vo et al., 2014*).

Deficits in goal-directed decision making, and specifically in how reward shapes selection of actions, are a core endophenotype shared across neuropsychiatric disorders, including schizophrenia, autism spectrum disorders (ASD), obsessive-compulsive disorder, and Tourette syndrome

(*Barch and Dowd, 2010*; *Corbett et al., 2009*; *Dichter et al., 2012*; *Dowd et al., 2016*; *Gillan and Robbins, 2014*; *Griffiths et al., 2014*; *Hill, 2004*; *Maia and Frank, 2011*; *Solomon et al., 2015*). In schizophrenia, impairments in action–outcome learning (*Gold et al., 2015*; *Morris et al., 2015*) may reflect perturbations to reinforcement learning error signals or the manner in which they are integrated to impact choice (*Hernaus et al., 2019*; *Hernaus et al., 2018*). Recent studies have also revealed reinforcement learning deficits in ASD patients (*Hill, 2004*; *Solomon et al., 2015*), with impaired choice accuracy driven by reduced win–stay choice patterns (*Solomon et al., 2015*).

Genetic association studies for neuropsychiatric disease have converged on synapses as key sites of disease pathophysiology (*DDD Study et al., 2014*; *Willsey et al., 2013*; *Willsey and State, 2015*). Neurexin1α (Nrxn1α) is an evolutionarily conserved synaptic adhesion molecule, for which rare de novo and inherited copy number variants confer significant risk for ASDs, schizophrenia, Tourette syndrome, and obsessive-compulsive disorder (*Ching et al., 2010*; *Duong et al., 2012*; *Huang et al., 2017*; *Kirov et al., 2009*; *Lowther et al., 2017*; *Rujescu et al., 2009*). The Neurexin family of proteins functions as a presynaptic hub for transynaptic binding of numerous postsynaptic partners at both excitatory and inhibitory synapses (*Missler et al., 2003*; *Südhof, 2017*). Consistent with their expression prior to synaptogenesis (*Harkin et al., 2017*; *Puschel and Betz, 1995*), Neurexins have been implicated in the initial specification and long-term integrity of synapses (*Anderson et al., 2015*; *Aoto et al., 2013*; *Chubykin et al., 2007*; *Krueger et al., 2012*; *Soler-Llavina et al., 2011*; *Südhof, 2017*; *Varoqueaux et al., 2006*). While *Nrxn1α* transcripts are broadly expressed throughout the brain, they are particularly enriched in cortico-striatal-thalamic loops proposed to govern motor control, action selection, and reinforcement learning (*Fuccillo et al., 2015*; *Ullrich et al., 1995*).

Behavioral abnormalities in Nrxn1α knockout animals include reduced nest building and social memory, increased aggression and grooming, enhanced rotarod learning, and male-specific reductions in operant responding under increasing variable interval responding schedules (*Dachtler et al., 2015*; *Esclassan et al., 2015*; *Etherton et al., 2009*; *Grayton et al., 2013*). Despite this broad dysfunction, the underlying mechanistic contributions of Nrxn1α to disease-relevant behaviors remain unclear, owing to our poor understanding of the specific computational algorithms and neural circuit implementations for the behavioral functions interrogated by these standard assays.

In this paper, we uncover widespread alterations in reward processing in Nrxn1α knockout mice, manifest as inefficient choice and altered control of task engagement. These deficits were observed across a range of value comparisons and feedback rates, suggestive of trait-like decision-making abnormalities. Modeling of choice patterns suggests these deficits are driven by impaired learning and representation of choice values. To reveal causal circuits for this reward processing defect, we performed brain region-specific deletion of Nrxn1α. We found that Nrxn1α disruption in excitatory telencephalic neurons, but not thalamic neurons recapitulated the choice and reward processing abnormalities of brain-wide Nrxn1α knockouts. Furthermore, telencephalic projection neuron-specific Nrxn1α disruption produced dysregulation of value-associated circuit activity prior to choice in direct pathway neurons of the dorsal striatum. Together, this work represents an important step in characterizing the genetic contributions to circuit dysfunction for a core neuropsychiatric disease-relevant behavior – how animals choose actions according to cost and benefit.

## Results

### Neurexin1α KOs have blunted responses to relative reward outcomes

We found that Nrxn1α knockout (KO) mice could perform basic light-guided operant responding with consistent task engagement (*Figure 1—figure supplement 1A–C*). Next, we specifically tested how Nrxn1α mutant mice use value information to guide future choice via a feedback-based paradigm (*Figure 1A*). Briefly, mice self-initiated consecutive two alternative forced-choice trials where each alternative was associated with contrasting reward volumes (12 µL versus either 0 µL, 2 µL, 6 µL, or 8 µL). To explore whether value comparisons were further influenced by reward scarcity, we tested four relative reward ratios in both high ($P_{rew} = 0.75$) and low ($P_{rew} = 0.4$) feedback regimes. Alternation of reward contingencies was used (triggered by 80% bias toward the larger reward in a moving 10-trial block) to maintain outcome sensitivity over hundreds of trials (*Figure 1A*; see *Alabi et al., 2019* and Materials and methods for further details). Performance in this task was

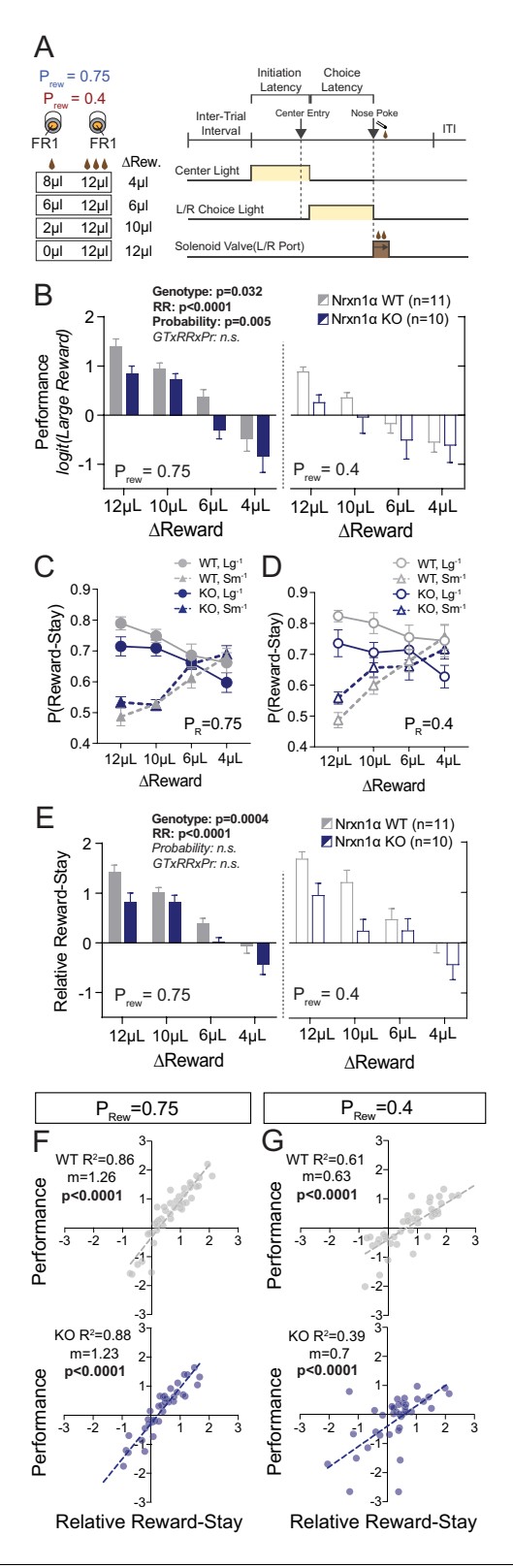

**Figure 1.** Neurexin1α disruption leads to deficits in value-based selection of actions. (**A**) Schematic of trial structure wherein mice perform repeated self-initiated trials with contrasting reward volumes associated with each port. Animals were tested at four relative reward ratios across high ($P_{rew}$ = 0.75) and low ($P_{rew}$ = 0.4) reinforcement rates. See Materials and methods for details. (**B**) Both probability of reinforcement and volume contrast modulate

*Figure 1 continued on next page*

*Figure 1 continued*

the probability at which mice select the large reward option. Nrxn1α KOs (blue, n = 10) select the high benefit alternative at a lower rate than their WT littermates (gray, n = 11) across reward environments (three-way RM ANOVA). (**C and D**) For both WT and KO animals, the relative magnitude of rewarded outcome has a significant effect on the stay-probability for that alternative. (**E**) The relative reward-stay (RRS), which quantifies the relative tendency of animals to repeat choices after specific outcomes, was sensitive to relative magnitude of rewards but not reward probability. In comparison to WT littermates, Nrxn1α KOs less dynamically alter their choice behavior after large reward outcomes than small reward outcomes (three-way RM ANOVA). (**F and G**) The RRS is a significant predictor of session performance for both WT and KO mice at both rates of reinforcement. Note RRS is a better predictor of task performance at high reinforcement rates, reflecting the preponderance of unrewarded outcomes in low reinforcement conditions. All data represented as mean ± SEM.

The online version of this article includes the following source data and figure supplement(s) for figure 1:

**Source data 1.** Source Data for *Figure 1*.
**Figure supplement 1.** Additional Behavioral Analyses in Nrxn1a KO mice.

---

significantly altered by the relative magnitude of rewarded outcomes for both wild-type and KO animals with larger reward contrasts driving more biased choice patterns (*Figure 1B*). Nonetheless, we observed a global decrease in session performance across relative reward contrasts in Nrxn1α KO mice as compared to wild type (*Figure 1B*), without genotypic differences in total reward consumed or task engagement (*Figure 2—figure supplement 1B and C*).

Performance could be altered by changes in: (1) how feedback is integrated over time; (2) sensitivity to outcome feedback; and (3) flexibility to changing contingencies (*Alabi et al., 2019*). To assess whether Nrxn1α KOs show altered influence of reward history on current choice, we employed logistic regression models to estimate the relative effects of choice and outcome (five preceding trials) on current choice (*Lau and Glimcher, 2005*; *Parker et al., 2016*; *Tai et al., 2012*). We found that wild-type mice and Nrxn1α KOs heavily discount all but the immediately preceding trial ($t-1$) (*Figure 1—figure supplement 1E–H*), suggesting a significant portion of choice variability can be accounted for by analyzing influences of the $t-1$ trial. We therefore calculated the relative reward-stay (RRS), a measure of the relative reinforcing properties of large versus small rewarded $t-1$ outcomes (previously relative action value in *Alabi et al., 2019*). We noted smaller gaps between large reward-stay and small reward-stay behavior in Nrxn1α KOs as compared with wild types (*Figure 1C and D*), leading to smaller RRS values across reward contrasts and feedback environment (*Figure 1E*). The significant correlation between RRS and performance across genotypes highlights the importance of outcome sensitivity on task performance (*Figure 1F and G*).

As deficits in behavioral adaptability have been observed across neuropsychiatric disorders and impact performance in this task (*Alabi et al., 2019*), we compared choice patterns at un-signaled contingency switches, noting no statistically significant alteration in KO mice (*Figure 1—figure supplement 1I*). We further probed cognitive flexibility with extra-dimensional set-shifting and spatial reversal tasks, again observing no performance differences between genotype (*Figure 1—figure supplement 1J and K*). In sum, choice abnormalities in Nrxn1α KO mice arise from decreased sensitivity to beneficial outcomes as opposed to altered feedback integration or impaired cognitive flexibility.

## Neurexin1α mutants exhibit abnormalities in outcome-related task engagement

The temporal relationship between action and reinforcement modulates the degree to which rewards shape behavior. To assess whether observed differences in outcome sensitivity resulted from divergent temporal patterns of performance, we compared task latencies. We observed no significant discrepancies in latency to initiate between Nrxn1α wild-type and KO mice across varied reward environments (*Figure 2A*), suggesting that observed outcome-associated choice is not attributable to global task disengagement. Recent evidence suggests local choice value can also modulate the vigor with which selected actions are performed (*Bari et al., 2019*; *Hamid et al., 2016*). If inefficient choice patterns of Nrxn1α KOs result from disrupted value encoding, we expect that the effects of recent outcomes on action vigor would be similarly blunted. To explore this, we compared outcome-dependent initiation latencies after large reward versus small reward outcomes

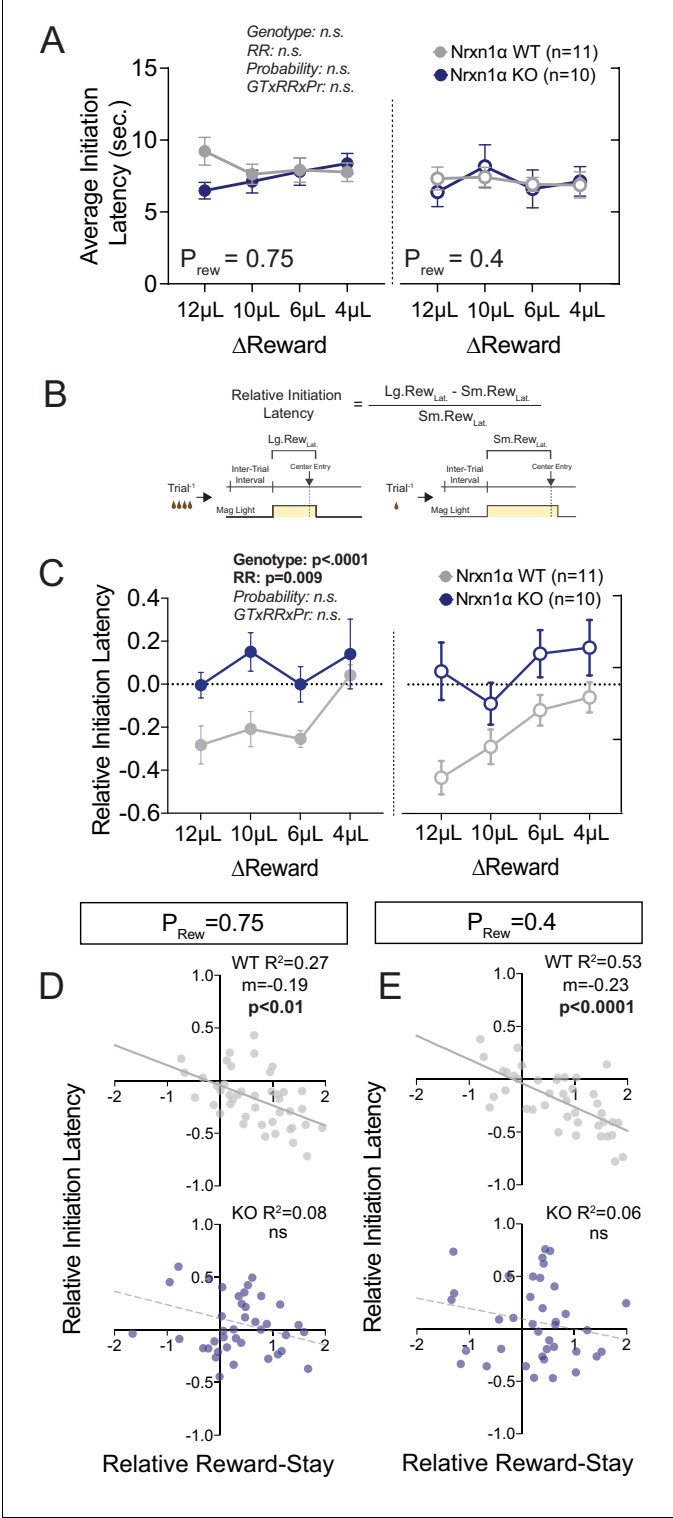

**Figure 2.** Neurexin1α mutants display altered outcome-dependent task engagement. (**A**) A proxy of task engagement was measured as the average latency from trial onset (center-light ON) to initiation. Nrxn1α KOs (blue, n = 10) do not exhibit global deficits in task engagement in comparison to WT animals (gray, n = 11) (three-way RM ANOVA). (**B**) Relative latency to initiate is a standardized comparison of initiation latencies following large rewarded outcomes and small rewarded outcomes within individual animals. (**C**) Nrxn1α WT mice modulate their trial-by-trial engagement in response to different rewarded outcomes, initiating trials more quickly after large reward outcomes than small reward outcomes. Nrxn1α KOs do not exhibit this outcome-sensitive modulation of

*Figure 2 continued on next page*

*Figure 2 continued*

task engagement (three-way RM ANOVA). (**D and E**, top) There is a significant relationship between the ability of WT mice to select actions in response to reward discrepancy (RRS) and their ability to upregulate task engagement (relative initiation latency) which is lost in KOs (**D and E**, bottom). All data represented as mean ± SEM.

The online version of this article includes the following source data and figure supplement(s) for figure 2:

**Source data 1.** Source data for *Figure 2*.
**Figure supplement 1.** Additional Task Latency and Reward Volume Data.

(*Figure 2B*). Interestingly, the relative latency to initiate trials in wild-type animals was significantly modulated by the relative reward ratio (*Figure 2C*, gray), with animals initiating trials more quickly after large reward outcomes than small reward outcomes. In contrast, Nrxn1α knockout mice were entirely unable to modulate initiation latency in response to the magnitude of previous reward (*Figure 2C*, blue). The strong inverse correlation between relative reward-stay and initiation latency was lost in Nrxn1α KO mice (*Figure 2D and E*). Thus, while there is no difference in average task latencies between wild types and KOs, Nrxn1α mutations disrupt outcome-modulated task engagement. We also observed a fixed elongation of choice latency in Nrxn1α mutants across reward environments (*Figure 2—figure supplement 1A*).

## Value processing abnormalities in the Neurexin1α mouse extend to cost-based decision making

To see whether choice behavior based on costs was similarly affected in Nrxn1α mutants, we associated two choice alternatives with distinct motor requirements (fixed ratio 3 [FR3] vs. FR1; *Figure 3A*). Reward contingencies in this paradigm were not alternated and after 75 trials of feedback, mice achieved a steady-state response pattern. Interestingly, Nrxn1α KO mice do not select low-effort alternatives as frequently as wild-type littermates, both during sampling and steady-state periods (*Figure 3B*). While we noted the KOs slowed more over the session (*Figure 3B*), no significant difference in steady-state task engagement was seen (*Figure 3C*). We continued to observe an effect of genotype on choice latency (*Figure 3D*) as in prior tasks.

## Reinforcement modeling reveals genotype-specific deficits in updating of outcome value

To uncover core decision-making processes underlying outcome-insensitive choice behavior in Nrxn1α mutants, we modeled action selection as a probabilistic choice between two alternatives with continually updating values (*Figure 4A*). We employed a modified Q-learning model with soft-max decision function, including five parameters: (1) learning rate ($\alpha$), which determines the extent to which new information about state-action pairing alters subsequent behavior; (2) reward compression parameter ($\gamma$), capturing the subjective benefit of a given reward volume; (3) inverse temperature parameter ($\beta$) ), linking the values of each option to choice output; (4) perseveration parameter ($\kappa$), capturing the effect of previous choices on subsequent choice, and (5) constant terms to capture spatial biases in choice behavior (see Materials and methods) (*Doya, 2007*; *Niv, 2009*; *Vo et al., 2014*).

We have previously demonstrated stable trait-like reward processing characteristics in this task (*Alabi et al., 2019*). In light of this, we grouped the choice data of individual animals across reward ratios to extract stable behavioral parameters. We fit our model using function minimization routines and found that it provided accurate predictions of individual animal choice patterns (*Figure 4B*). Fitting choice data for wild-type and KO mice, we demonstrated that Nrxn1α KO mice have significantly lower $\alpha$ and $\gamma$ parameters (*Figure 4C and D*), suggesting a global deficit in the updating and representation of choice values guiding decisions (*Figure 4E*). In contrast, we did not observe genotypic differences for the $\beta$, $\kappa$, or bias parameters (*Figure 4F and G* and *Figure 4—figure supplement 1*), suggesting no systemic differences in how the genotypes transform value representations into actions (*Figure 4H*).

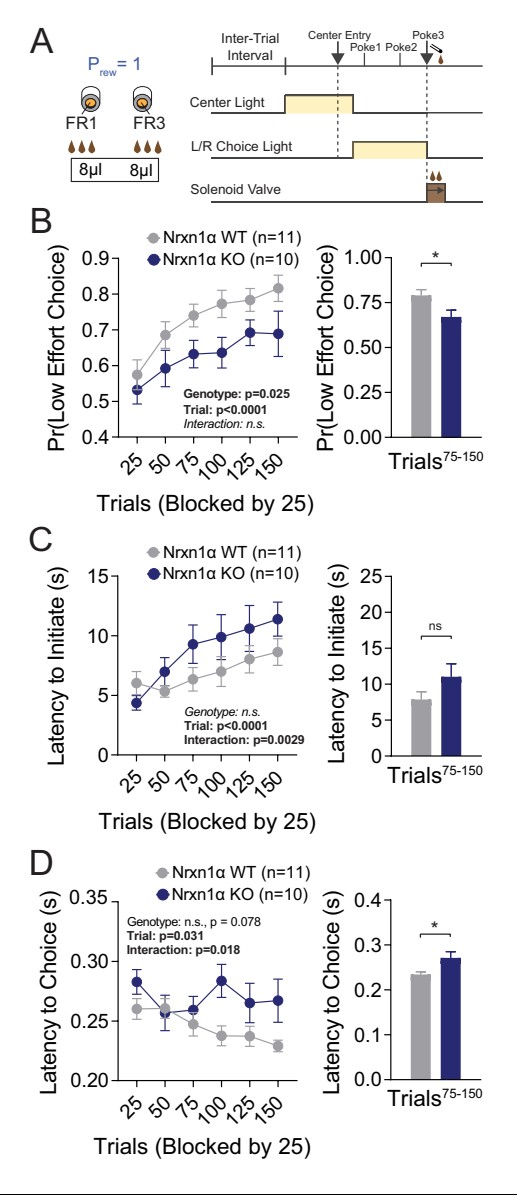

**Figure 3.** Neurexin1α mutants display a deficit in the selection of actions based on costs. (**A**) Effort paradigm schematic. Mice distribute choices in a session with fixed contingency lasting 150 trials. Animals were given choices with equal reward outcomes, but different effort requirements (FR3 vs. FR1). (**B**) Nrxn1α KOs (blue, n = 10) choose less costly alternatives at a lower rate than their WT littermates (gray, n = 11) (two-way RM ANOVA). The distribution of choice in both WT and KO mice is altered over the course of the block as mice acquire information about the reward contingency, with a stable difference observed over the final 75 trials (two-sample t-test *p=0.023). (**C**) Nrxn1α KOs exhibited a clear interaction between trial and latency to initiate, slowing as they performed more high effort trials (two-way RM ANOVA). Nevertheless, there was no statistically significant difference in engagement at steady state (two-sample t-test p=0.14). (**D**) The longer
*Figure 3 continued on next page*

## Ablation of Neurexin1α in telencephalic projection neurons recapitulates value-based abnormalities

We next sought to identify molecularly causal circuits relevant for the deficits in value updating exhibited by Nrxn1α KO mice. Multiple telencephalic excitatory regions, which exhibit high expression of *Nrxn1α* mRNA, have been implicated in the regulation of action–outcome association and encoding of subjective choice value (*Bari et al., 2019*; *Euston et al., 2012*; *Noonan et al., 2011*; *Padoa-Schioppa and Conen, 2017*; *Rushworth et al., 2011*; *Rushworth et al., 2012*). To test whether Nrxn1α loss-of-function in these circuits could drive reward processing deficits, we crossed a Neurexin1α conditional allele (*Nrxn1α^{fl}*), where exon 9 is surrounded by loxP sites, to the Nex-Cre transgenic line, where Cre-recombinase is driven from the *Neurod6* locus in postmitotic progenitors of cortical, hippocampal, and amygdalar projection neurons (*Goebbels et al., 2006*; *Figure 5A and B*). mRNA from cortical dissection of *Nrxn1α^{fl/fl}*; *Nex^{Cre/+}* revealed a 3.5× decrease in *Nrxn1α* transcripts spanning exon 9 as compared to *Nrxn1α^{fl/fl}*; *Nex^{+/+}* (*Figure 5C*, left), and a modest degree of nonsense-mediated decay with a downstream probe (*Figure 5C*, right). Given the early expression of Cre from the Nex-^{Cre/+} line, it is likely that the *Nrxn1α^{fl}* allele is recombined prior to its endogenous expression (*Lukacsovich et al., 2019*). We choose this early deletion so as to best model the pathophysiological processes secondary to Nrxn1α mutations and make direct comparison to the phenotypes observed in the constitutive Nrxn1α KO mice.

In order to test the effects of Nrxn1α loss-of-function in telencephalic projection neurons, we repeated the value-based tasks in *Nrxn1α^{fl/fl}*; *Nex^{Cre/+}* mice. To account for potential hypomorphic effects of the Nrxn1α conditional allele as well as effects of constitutive Cre expression in the Nex^{Cre} line, we utilized two controls: *Nrxn1α^{+/+}*; *Nex^{Cre/+}* and *Nrxn1α^{fl/fl}*; *Nex^{+/+}*. We observed a significant effect of Nex^{Cre} deletion of Nrxn1α on relative reward stay as compared to both control groups (*Figure 5D*). Similar to global Nrxn1α deletion, *Nrxn1α^{fl/fl}*; *Nex^{Cre/+}* mutant animals were less able to bias their choice patterns toward more beneficial outcomes. We noted no consistent difference in behavioral flexibility in these mice (*Figure 5—figure supplement 1A*). Neither the *Nrxn1α^{fl/fl}*; *Nex^{+/+}* conditional control nor the *Nrxn1α^{fl/fl}*; *Nex^{Cre/+}*

*Figure 3 continued*

choice latencies previously described in Nrxn1α KOs was observed in steady-state responding (two-way RM ANOVA; two-sample t-test *p=0.017). All data represented as mean ± SEM.

The online version of this article includes the following source data for figure 3:

**Source data 1.** Source Data for *Figure 3*.

mutant animals displayed the reward-related modulation of initiation latencies observed in the Nrxn1α wild-type animals (*Figure 5—figure supplement 1B*), precluding conclusions regarding local modulation of action vigor. Similar to constitutive Nrxn1α KOs, we noted an increased choice latency across varied reward environments (*Figure 5—figure supplement 1C*). To test whether deficits in working memory could contribute to our choice phenotype, we assessed spontaneous alternation behavior of $Nrxn1\alpha^{fl/fl}$; $Nex^{Cre/+}$ and $Nrxn1\alpha^{fl/fl}$; $Nex^{+/+}$ conditional control littermates, observing no genotypic differences (*Figure 5—figure supplement 1G*).

To assess whether forebrain-specific Nrxn1α KOs generated similar reward processing abnormalities as Nrxn1α constitutive KOs, we again employed reinforcement modeling of choice data. As in whole-brain Nrxn1α KOs, we observed a significant effect of genotype on learning rate and reward discrimination parameters (*Figure 5E and F*), generating a leftward shift in the distribution of action value contrasts in $Nrxn1\alpha^{fl/fl}$; $Nex^{Cre/+}$ mice (*Figure 5G*). In keeping with prior data, we observed no genotypic differences in value-related explore/exploit behavior, choice persistence, or average bias (*Figure 5H–J* and *Figure 5—figure supplement 1D*). In our effort-based cost paradigm, the $Nrxn1\alpha^{fl/fl}$; $Nex^{Cre/+}$ conditional mutants exhibited reduced selection of the lower-cost alternative than both groups of control animals (*Figure 5K*). Average task engagement was not abnormal in these animals (*Figure 5L* and *Figure 5—figure supplement 1E*), but we again noted a persistent increase in choice latency (*Figure 5M* and *Figure 5—figure supplement 1F*). Together, these data suggest that embryonic deletion of Nrxn1α in telencephalic excitatory neurons is sufficient to produce similar perturbations of reward processing and choice as those observed in whole-brain Nrxn1α KO mice.

## Deletion of Neurexin1α in thalamic nuclei does not recapitulate choice deficits

Neurexin1α is highly expressed in multiple subcortical regions involved in the selection and performance of goal-directed actions (*Bradfield et al., 2013*; *Díaz-Hernández et al., 2018*; *Fuccillo et al., 2015*; *Ullrich et al., 1995*). In order to assess the specificity of telencephalic excitatory Nrxn1α conditional KO (cKO) in driving reward processing abnormalities, we conditionally deleted Nrxn1α in developing thalamic nuclei via an Olig3-Cre driver line (*Figure 6A–C*). In contrast to telencephalic excitatory cKO, thalamic cKO could not recapitulate the deficits in value processing observed in whole-brain Nrxn1α mutants (*Figure 6D* and *Figure 6—figure supplement 1A–C*). There was no significant genotypic difference in the ability to modulate choice distributions in response to reward (*Figure 6D*), nor in any parameters of the fitted reinforcement model (*Figure 6E–J* and *Figure 6—figure supplement 1D*). Additionally, we noted no significant genotypic differences in choice allocation away from effortful alternatives (*Figure 6K–M* and *Figure 6—figure supplement 1E*). The only aspect of the constitutive KO phenotype partially recapitulated by the thalamic cKOs was increased choice latency in the fixed contingency paradigm (*Figure 6—figure supplement 1F*, but see *Figure 6M*).

## Characterizing value-modulated neural signals within dorsal striatum

Our data suggest that both global and telencephalic excitatory neuron-specific Nrxn1α mutants exhibit inefficient choice patterns secondary to deficits in value encoding/updating. Given the function of Nrxn1α in supporting excitatory synaptic transmission in hippocampal circuits (*Etherton et al., 2009*), we explored how its disruption might impact neural activity within key reinforcement learning circuits. We focused on direct pathway spiny projection neurons (dSPNs) of the dorsal striatum, as this population: (1) is a common downstream target of forebrain excitatory populations that both encode value and express Nrxn1α in their presynaptic terminals (*Bari et al., 2019*; *Bradfield et al., 2013*; *Parker et al., 2019*); (2) encodes reward values (*Donahue et al., 2019*; *Samejima et al., 2005*; *Shin et al., 2018*); and (3) can bias choice in value-based operant tasks

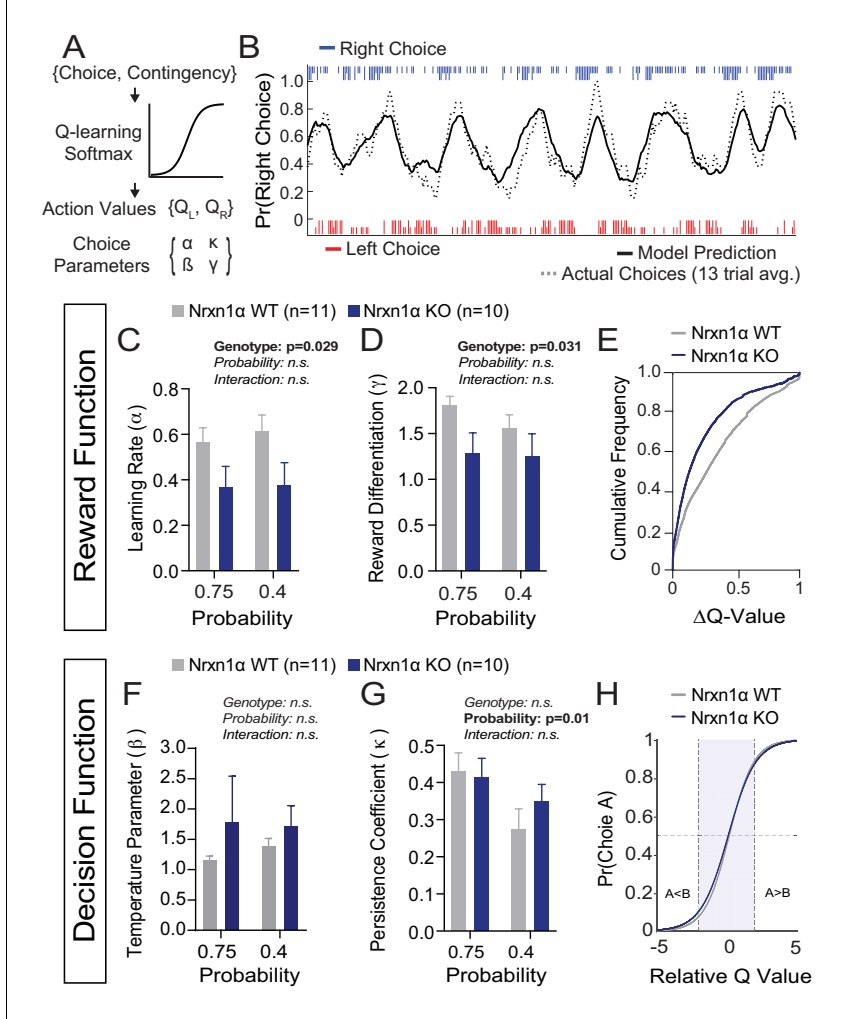

**Figure 4.** A deficit in value updating underlies abnormal allocation of choices in Neurexin1α mutants. (**A**) Q-learning reinforcement model. Mouse choice was modeled as a probabilistic choice between two options of different value ($Q_L$,$Q_R$) using a softmax decision function. Data from each reinforcement rate were grouped before model fitting. (**B**) Example of model prediction versus actual animal choice. Choice probability calculated in moving window of 13 trials. Long and short markers indicate large and small reward outcomes. (**C and D**) As compared to littermate controls (gray, n = 11), Nrxn1α mutants (blue, n = 10) exhibit a deficit in the learning rate, α, which describes the weight given to new reward information and γ, a utility function that relates how sensitively mice integrate rewards of different magnitudes (two-way RM ANOVA). (**E**) Nrxn1α KOs exhibit an enrichment of low ΔQ-value trials. (**F and G**) Nrxn1α mutants do not exhibit significant differences in explore–exploit behavior (**F**, captured by β) or in their persistence toward previously selected actions (**G**, captured by κ). (**K**) There is no significant difference in the decision function of Nrxn1α wild-type and mutant animals. All data represented as mean ± SEM. Bias figures can be found in *Figure 4—figure supplement 1*.

The online version of this article includes the following source data and figure supplement(s) for figure 4:

**Source data 1.** Source Data for *Figure 4*.

**Figure supplement 1.** Additional Reinforcement Learning Model Parameters.

---

(*Kravitz et al., 2012*; *Tai et al., 2012*). To select for striatal dSPNs, we expressed GCamp6f in neurons projecting to the substantia nigra reticulata (SNr), via combined injection of retroAAV2. EF1α−3xFLAG-Cre in the SNr and AAV5.hSyn-DIO-GCamp6f in the dorsal striatum of control NEX-$^{Cre}$ mice (*Figure 7A and B*). Putative direct pathway SPNs (p-dSPNs) exhibited reproducible Ca$^{2+}$ activity patterns in relation to three task epochs – trial start (center port light on), self-initiation (center port entrance), and choice/reward delivery (side port entry) (*Figure 7C*), despite exhibiting

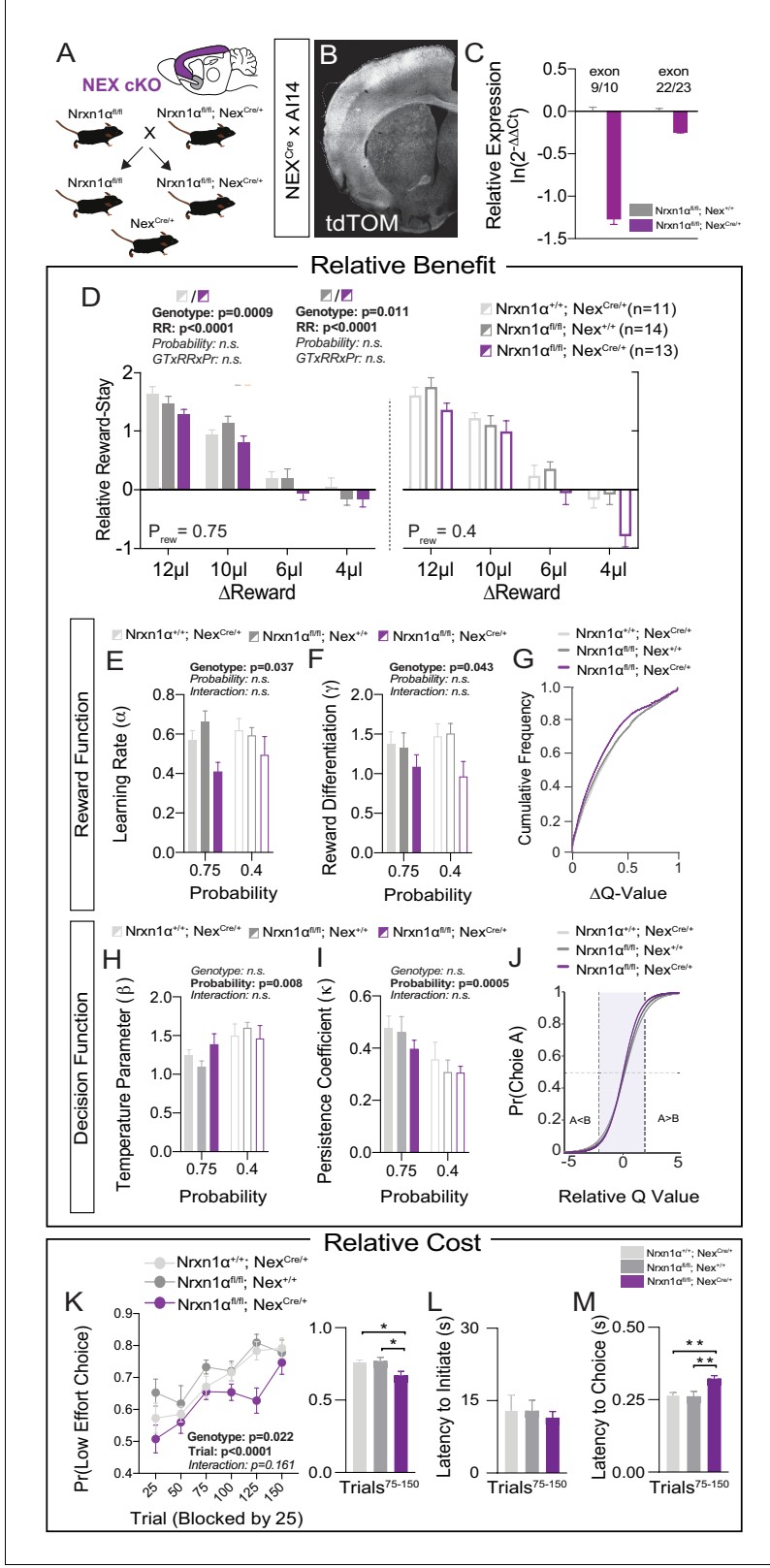

**Figure 5.** Restricted telencephalic excitatory neuron deletion of Neurexin1α recapitulates choice abnormalities of constitutive KO. (**A**) Nrxn1α was conditionally inactivated in telencephalic excitatory neurons by crossing a Nrxn1α-conditional knockout allele onto Nex[Cre] line. Controls for both the Nex (light gray) and Neurexin1α-conditional (dark gray) allele were analyzed. (**B**) Coronal section of brain from NEX[Cre/+];Ai14 (LSL-tdTOM) reporter cross

*Figure 5 continued on next page*

*Figure 5 continued*

showing restriction of tdTOM fluorescence to cortex, hippocampus, and a subdomain of the amygdala. (C) RT-qPCR of RNA from adult mouse cortex (n = 3 for Nrxn1α$^{fl/fl}$; Nex$^{+/+}$[dark gray] and Nrxn1α$^{fl/fl}$; Nex$^{Cre/+}$[purple]). Cre-mediated recombination results in reduced expression of Nrxn1α mRNA detected by exon 9 probe (two-sample t-test: p<0.0001) and moderate nonsense-mediated decay (two-sample t-test: p<0.01). (D) Nrxn1α$^{fl/fl}$; Nex$^{Cre/+}$ mutant animals (purple; n = 13) exhibit a reduction in relative reward-stay as compared with Nrxn1α$^{fl/fl}$; Nex$^{+/+}$(dark gray; n = 14) and Nrxn1α$^{+/+}$; Nex$^{Cre/+}$ (light gray; n = 11) controls. No difference in choice allocation was observed between control animals (genotype: p=0.88, relative reward: ***p<0.0001, probability: p=0.26, three-way interaction: p=0.25; three-way RM ANOVA). (E and F) Similar to Nrxn1α constitutive knockouts, Nrxn1α$^{fl/fl}$; Nex$^{Cre/+}$ mutant mice have a deficit in utilizing new reward information to update and represent choice values. The mutants exhibit a deficit in the learning rate (α) and in the reward volume sensitivity parameter (γ) (both analyzed by two-way RM ANOVA). (G) This leads to an enrichment of low ΔQ-value trials in mutant mice. (H–J) Nrxn1α$^{fl/fl}$; Nex$^{Cre/+}$ mutants do not differ from littermate controls for the relationship between choice value and decision behavior (H) and biases toward previous choice behavior (I). As a result, there is no significant difference in the decision function of control and mutant animals. (K–M) Nrxn1α$^{fl/fl}$; Nex$^{Cre/+}$ mutants exhibit a deficit in the allocation of choices guided by relative choice costs (K, two-way RM ANOVA, left; one-way ANOVA w/Tukey's multiple comparison, right, *p<0.05). Mutants exhibit no difference in task engagement (L, one-way ANOVA w/ Tukey's multiple comparison, p>0.05) but recapitulate deficit in choice latencies (M, one-way ANOVA w/Tukey's multiple comparison, **p<0.01). All data represented as mean ± SEM.

The online version of this article includes the following source data and figure supplement(s) for figure 5:

**Source data 1.** Source Data for *Figure 5*.

**Figure supplement 1.** Additional Behavioral Analysis of Telencephalic Excitatory Neuron Nrx1a Deletion.

---

smaller average signals than during task disengagement (*Figure 7—figure supplement 1A*). The lack of similar waveforms on the isosbestic 405 nm channel confirms the specificity of these epoch-aligned Ca$^{2+}$ signals (*Figure 7—figure supplement 1B*).

Recent population Ca$^{2+}$ imaging of striatal SPN populations has revealed a prolonged ramping activity prior to action sequence initiation (*London et al., 2018*). Given our data (*Figure 2*) and other work documenting the modulation of initiation latency by prior outcome (*Bari et al., 2019*), in addition to the technical challenges of reliably separating the choice and outcome components of the Ca$^{2+}$ waveform (*Figure 7—figure supplement 1C–F*), we investigated the preinitiation window as a key epoch for value-modulated signals in striatal direct pathway neurons. An average of all trials aligned by initiation demonstrated slow and fast phases of the p-dSPN Ca$^{2+}$ waveform (*Figure 7D*). To understand how reward correlates with wild-type p-dSPN activity, we segregated trials by previous (t−1) outcome. We found that most pre-initiation epochs following a 'small reward' trial had elevated activity compared to the population Ca$^{2+}$ average, while trials following 'large reward' had suppressed activity relative to the population average (*Figure 7E*), a trend similarly present in the population data (*Figure 7F*). To further quantify signal dynamics, we examined the slow ramping phase, occurring ~10 s before an initiation, and the fast peaking phase, occurring 1 s before initiation. We found that both signal components were differentially modulated by reward outcome: (1) for slow ramping, (t−1) large reward outcomes result in negative ramping or silencing of p-dSPN activity in comparison with small rewards (*Figure 7G and H*); (2) for fast peaking, larger rewards result in steeper peak activity as compared to smaller rewards (*Figure 7J and K*). Furthermore, we noted significant correlations between both measures and trial-by-trial comparative action values (*Figure 7I and L*; see Materials and methods), suggesting these p-dSPN signals may reflect value information employed for future action selection.

## Neurexin1α deletion in excitatory telencephalic projection neurons disrupts value-associated striatal neuron activity

To examine whether deletion of Nrxn1α from telencephalic projection neurons disrupted value-modulated neural signals within striatum, we performed population Ca$^{2+}$ imaging of p-dSPNs in both Nrxn1α$^{+/+}$; Nex$^{Cre/+}$ (Nex-Control) and Nrxn1α$^{fl/fl}$; Nex$^{Cre/+}$ (Nex-Nrxn1α$^{cKO}$) mice during our serial reversal task. While we did not uncover a difference for the slow ramp signal component between genotypes (*Figure 8A–C*), we found that the slope of the fast peak was consistently lower in Nex-Nrxn1α$^{cKO}$ (*Figure 8D and E*). Furthermore, this deficit was specifically associated with failure to increase peak activity in response to large reward volumes (*Figure 8F and G*). To assure that our

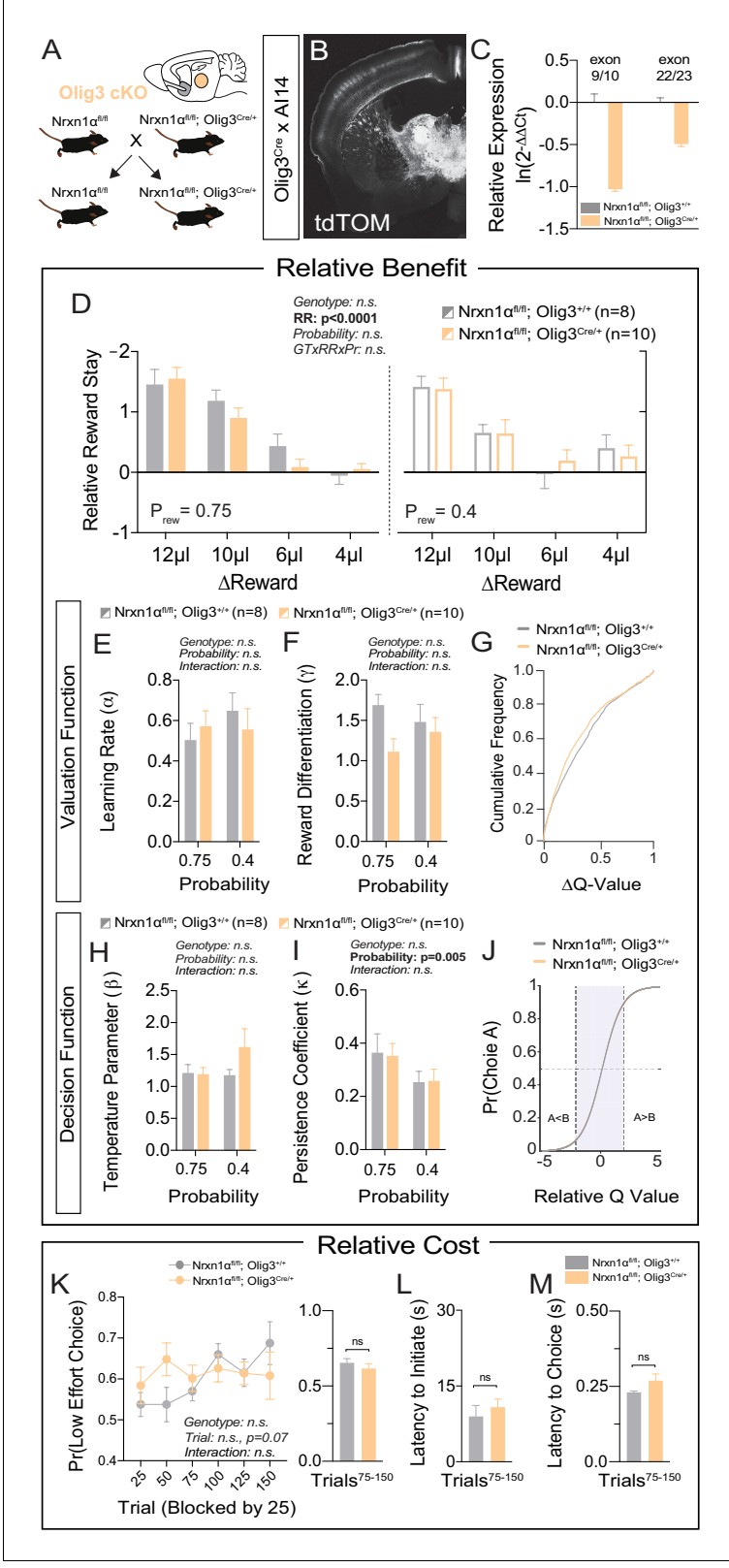

**Figure 6.** Specific deletion of Neurexin1α in thalamic nuclei does not reproduce choice abnormalities observed in constitutive KO. (**A**) Neurexin1α was conditionally inactivated in thalamic progenitor cells by crossing the Neurexin1α-conditional knockout line onto the Olig3-Cre line. (**B**) Coronal section of Olig3[Cre]; Ai14 reporter cross showing expression of tdTOM broadly throughout thalamic nuclei. (**C**) RT-qPCR of RNA from adult mouse

*Figure 6 continued on next page*

*Figure 6 continued*

thalamus (n = 2 for Nrxn1$\alpha^{fl/fl}$;Olig3$^{+/+}$ (gray); n = 3 for Nrxn1$\alpha^{fl/fl}$;Olig3$^{Cre/+}$(orange)). Cre-mediated recombination results in reduced expression of Nrxn1$\alpha$ mRNA detected by exon 9 probe (two-sample t-test: p<0.0001) and moderate nonsense-mediated decay (two-sample t-test: p<0.001) (**D**) Nrxn1$\alpha^{fl/fl}$;Olig3$^{Cre/+}$ mutant animals (orange; n = 10) do not exhibit changes in relative reward-stay in comparison with Nrxn1$\alpha^{fl/fl}$;Olig3$^{+/+}$(gray; n = 8) control animals. (**E–G**) Nrxn1$\alpha^{fl/fl}$;Olig3$^{Cre/+}$ mutant mice do not have a deficit in updating or representing choice values (two-way RM ANOVA). (**H–J**) Nrxn1$\alpha^{fl/fl}$;Olig3$^{Cre/+}$ mutants exhibit a normal relationship between choice values and decision behavior. (**K–M**) Nrxn1$\alpha^{fl/fl}$;Olig3$^{Cre/+}$ mutants do not exhibit a deficit in the allocation of choices guided by relative choice costs (**K**, two-way RM ANOVA, left; two-sample t-test, right, p>0.05). Mutants exhibit no difference in task engagement (**L**, p>0.05) or in choice latencies (**M**, p>0.05). All data represented as mean ± SEM.

The online version of this article includes the following source data and figure supplement(s) for figure 6:

**Source data 1.** Source Data for *Figure 6*.
**Figure supplement 1.** Additional behavioral analysis of thalamic neuron Nrx1a deletion.

strategy for labeling d-SPNs, wherein Cre becomes expressed in the recorded spiny neurons, did not alter recurrent inhibition, we compared a separate set of Nex-Nrxn1$\alpha^{cKO}$ mice injected with either retroAAV2.EF1$\alpha$−3xFLAG-Cre or retroAAV2.hSyn-GFP-$\Delta$Cre (an enzymatically inactive truncated version of Cre) in the SNr and noted no difference in the frequency or amplitude of miniature inhibitory postsynaptic currents (mIPSCs) according to virus (*Figure 8—figure supplement 1A and B*). To rule-out any potential effects on excitatory striatal afferents, we performed a similar experiment on Nrxn1$\alpha^{fl/fl}$ mice, again noting no difference in the miniature excitatory postsynaptic currents (mEPSCs) in retrograde Cre versus $\Delta$Cre viruses (*Figure 8—figure supplement 1C and D*). Together, these data suggest that telencephalic excitatory neuron-specific Nrxn1$\alpha$ mutants do not exhibit global disruptions of striatal circuit dynamics, but a specific outcome-associated perturbation in fast peak activity prior to trial initiation.

To better understand whether mutation-associated changes in striatal neural signals are related to specific components of value-based decision making, we developed a linear-mixed effects model to explain variability in the preinitiation phases of p-dSPN signals. Our model included variables for reward processing (prior trial reward outcome and reward prediction error, disparity in action value between choices in the upcoming trial), choice behavior (choice, explore–exploit, and stay–shift strategies), task engagement (initiation latencies), and lagging regressors to reflect 'carry-over' effects from previous trials (*Figure 8H*, see Materials and methods). We found that blunting of fast peak dynamics in Nex-Nrxn1$\alpha^{cKO}$ mutants was specific to aspects of reward processing – that is, while peak slopes had significant correlation to reward history, reward prediction error, and comparative choice values in wild-type mice, these outcome-sensitive signal components were absent in mutant striatal population dynamics (*Figure 8H*). In contrast, value-modulated signal components are preserved in the mutants during slow ramping (*Figure 8H*), supporting a circumscribed alteration in striatal value coding. Together, these data demonstrate disrupted reward responsive activity in direct pathway SPNs upon ablation of Nrxn1$\alpha$ in a subset of excitatory forebrain neurons. These changes are broadly consistent with our behavioral analysis showing Nrxn1$\alpha$ knockout in frontal projection neurons produced lower learning rate and sensitivity to outcome magnitudes (*Figure 5E and F*), generating smaller Q value discrepancies (*Figure 5G*).

## Discussion

Understanding genetic contributions to brain disease requires bridging the sizeable chasm between molecular dysfunction and behavioral change. While behaviorally circumscribed neural circuits provide a logical intermediary substrate, it has been challenging to identify disease-relevant neural populations owing to: (1) difficulty in finding assays that provide stable readouts of relevant behavioral constructs; (2) incomplete understanding of specific computational algorithms and neural circuit implementations for behavioral constructs; (3) challenges localizing relevant neural circuits wherein

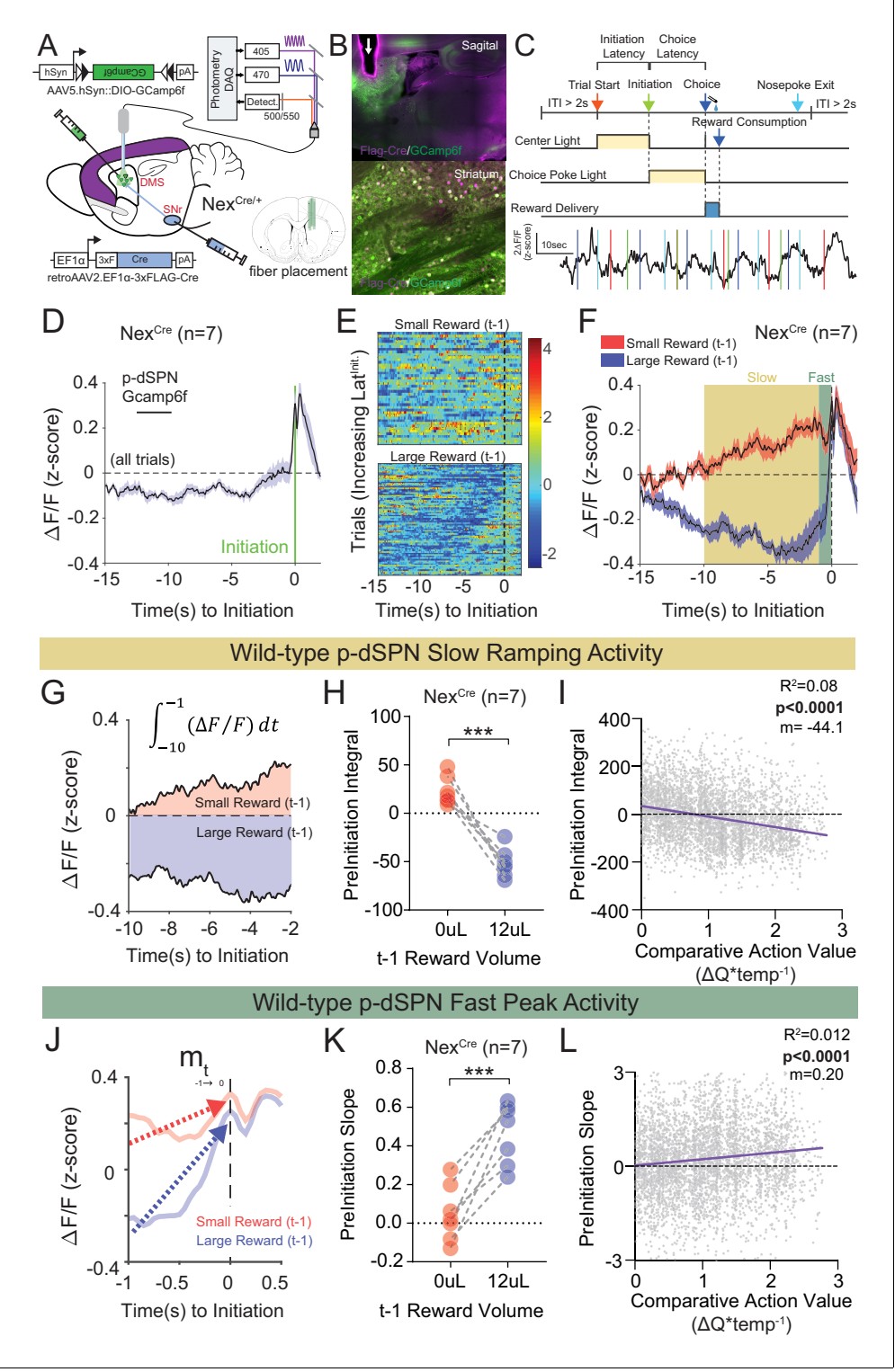

**Figure 7.** Quantifying value correlates in putative direct pathway SPNs of the dorsomedial striatum. (**A**) Schematic of experimental scheme. Control (Nrxn1α[+/+]; Nex[Cre/+], n = 7) mice were injected with a retro-AAV2-EF1α−3xFLAG-Cre virus in the substantia nigra, pars reticulata (SNr). Ipsilateral injection of Cre-dependent GCamp6f allowed for enrichment of putative direct pathway SPNs (p-dSPNs). (**B**, top) Sagittal section of Nex[Cre] brain showing GCamp6f expression in dorsal striatal SPNs and placement of 400 μm optic fiber (white arrow). (**B**, bottom) Magnified view of striatum showing colocalization of nuclear FLAG-Cre and cytoplasmic GCamp6f. (**B**, bottom left) Location of fiber placements in Nex[Cre/+]. (**C**, top) Trial schematic and relationship of specific task

*Figure 7 continued on next page*

*Figure 7 continued*

epochs with p-dSPN $Ca^{2+}$ signal (bottom). (**D**) Peristimulus time histogram (PSTH) of $\Delta F/F$ for $Nex^{Cre/+}$ aligned to initiation event (all trials). The initiation of the action sequence (green bar) is associated with a rise in p-dSPNs activity. (**E**) Representative heat map of individual animal trials segregated by reward outcome on ($t-1$) trial (sorted by the latency to initiate). Trials following a large reward have greater signal suppression than those following small reward. (**F**) PSTH of $\Delta F/F$ for $Nex^{Cre/+}$ aligned to initiation event (segregated by outcome on ($t-1$)). Preinitiation of p-dSPN dynamics exhibits two components – a slow ramping phase (yellow, $time_{-10\rightarrow-1}$) followed by a fast spike phase (green, $time_{-1\rightarrow init}$), both of which are modulated by ($t-1$) reward outcome. (**G**) The slow ramping phase is quantified by the integral of GCamp signal $-10$ s to $-1$ s before initiation. (**H**) There is a significant effect of ($t-1$) reward volume on the preinitiation integral during slow ramping with large rewards showing greater silencing of p-dSPN activity (paired t-test, ***p=0.0002). (**I**) Preinitiation integral inversely correlates with the comparative action value of the upcoming trial, which is calculated using probability estimates from fitted reinforcement learning models and reflects the disparity in choice value on a trial to trial basis. (**J**) The dynamics of the fast peak phase are represented by the average slope of GCamp signal from $-1$ s till initiation. (**K**) There is a significant effect of ($t-1$) reward volume on preinitiation slope during the fast peak phase (paired t-test, ***p=0.0006) with large rewards showing steeper subsequent preinitiation slopes. (**L**) Preinitiation slope positively correlates with the comparative action value of the upcoming trial.

The online version of this article includes the following source data and figure supplement(s) for figure 7:

**Source data 1.** Source Data for *Figure 7*.
**Figure supplement 1.** Additional Photometry Analyses in Wildtype and Mutant Mice.

gene perturbations drive behavioral dysfunction; and (4) limitations in correlating mutation-associated patterns of neural activity with abnormal execution of behavior.

Here we addressed these obstacles while investigating value-processing deficits in mice harboring mutations in Nrxn1α, a synaptic adhesion molecule associated with numerous neuropsychiatric disorders (*Dachtler et al., 2015*; *Duong et al., 2012*; *Huang et al., 2017*; *Kirov et al., 2009*; *Rujescu et al., 2009*; *Sanders et al., 2015*; *Südhof, 2008*). We found that constitutive Nrxn1α KO mice exhibited reduced bias toward more beneficial outcomes (modeled by greater reward volumes) and away from more costly actions (modeled by higher response schedules). Reinforcement modeling of choice behavior suggested altered mutant decision making resulted from deficits in the updating and representation of choice value as opposed to how these values are transformed into action. Using brain region-specific gene manipulation, we demonstrated that deletion of Nrxn1α from telencephalic projection neurons, but not thalamic neurons, was able to recapitulate most aspects of the reward processing deficits observed in constitutive Nrxn1α KOs. Finally, we investigated how circuit-specific Nrxn1α mutants altered value-modulated neural signals within direct pathway neurons of the dorsal striatum. We found that while fast peak $Ca^{2+}$ activity immediately preceding trial initiation strongly reflected aspects of prior and current action values in wild-type mice, value-coding signals were disrupted in telencephalic-specific Nrxn1α mutants.

## Deficits in value-based action selection in Neurexin1α mutants

Reframing the study of disease-associated behaviors into endophenotypes is a powerful approach to revealing underlying genetic causality. Nevertheless, the study of disease-relevant cognitive endophenotypes in mice has proven challenging. Here we employed a feedback-based, two-alternative forced choice task that forces value comparisons between choices of differing reward magnitude and required effort. We believe this task has many advantages for investigating cognitive dysfunction associated with neuropsychiatric disease risk genes such as Nrxn1α. First, we have previously shown that it produces stable within-mouse measures of benefit and cost sensitivity (*Alabi et al., 2019*), ideal for revealing between-genotype differences. Second, it probes how outcome value is used to direct future action selection – a core neural process perturbed across many of the brain disorders in which Nrxn1α mutations have been implicated (*Dichter et al., 2012*; *Gillan and Robbins, 2014*; *Maia and Frank, 2011*).

We find that global deletion of Nrxn1α resulted in a persistent deficit in outcome-associated choice allocation, driven strongly by reductions in win–stay behavior (*Figure 1C–E*). Interestingly,

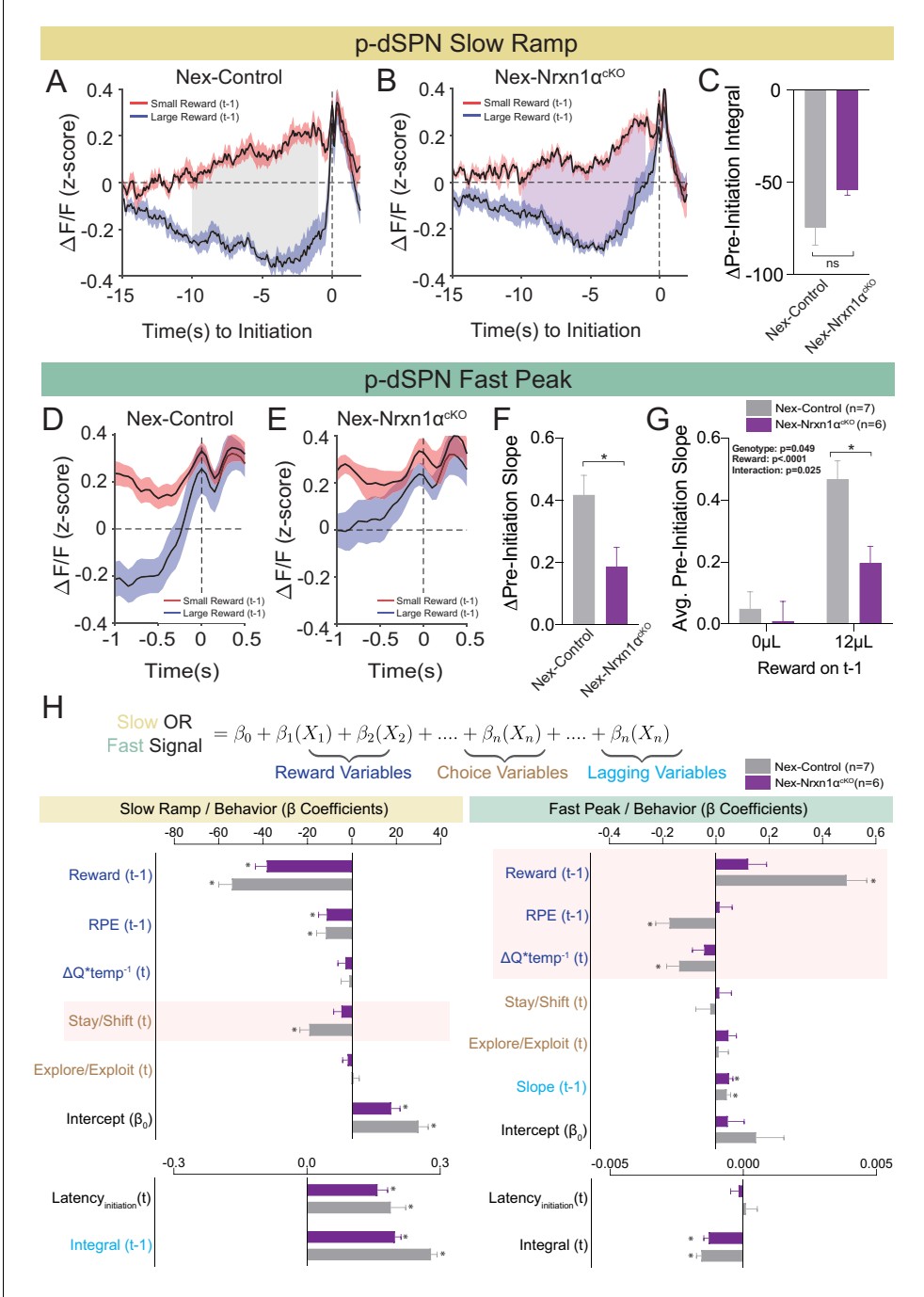

**Figure 8.** Restricted telencephalic excitatory neuron deletion of Neurexin1α produces a deficit in fast peak activity in p-dSPNs of the DMS. (**A and B**) PSTH of ΔF/F for Nex-control (Nrxn1α$^{+/+}$; Nex$^{Cre/+}$, n = 7, gray) and Nex-Nrxn1α$^{cKO}$ (Nrxn1α$^{fl/fl}$; Nex$^{Cre/+}$, n = 6, purple) mice, respectively, aligned to initiation event (segregated by outcome on $t$−1). Shaded region corresponds to the difference in the preinitiation integral following large and small reward outcomes. (**C**) There is no statistically significant difference between Nex-control and Nex-Nrxn1α$^{cKO}$ in the Δpre-initiation integral of large versus small rewards (two-sample t-test, n.s., p=0.084). (**D and E**) PSTH of ΔF/F for control and mutant animals, respectively, in the fast peak phase of preinitiation activity. (**F**) Nex-Nrxn1α$^{cKO}$ exhibit smaller disparity in fast peak signals after unique reward outcomes, as evidenced by significant effect of genotype on Δpre-initiation slope of the fast peak (two-sample t-test, *p=0.025). (**G**) This difference in Δpre-initiation slope arises from a blunted GCamp response in mutants to large reward outcomes (two-way RM ANOVA). (**H**) Modeling Ca$^{2+}$ signal dynamics as function of reward variables (blue), prior/future choice (gold), and lagging regressors (light blue) to capture prior circuit states. Value modulation of fast peak activity is blunted in

*Figure 8 continued on next page*

*Figure 8 continued*

Nex-Nrxn1α$^{cKO}$ mice (highlighted red box), while other components of the signal remain intact. Slow ramping is largely intact in mutant animals. All data represented as mean ± SEM.

The online version of this article includes the following source data and figure supplement(s) for figure 8:

**Source data 1.** Source Data for *Figure 8*.

**Figure supplement 1.** Retrograde labeling strategy does not alter excitatory or inhibitory basal synaptic transmission.

similar reductions in win–stay behavior during feedback-based tasks have been demonstrated to drive choice inefficiency in both schizophrenia (*Saperia et al., 2019*) and autism (*Solomon et al., 2015*), disorders for which Nrxn1α has been implicated. We observed that this value-based dysregulation manifests not only for the selection of higher-benefit actions, but also in the selection of less costly choices (*Figure 3*), as well as in the outcome-dependent modulation of task engagement as read out by initiation latency (*Figure 2*). Together, these data converge to suggest Nrxn1α mutations disrupt the function of brain circuits that internally represent value or circuits that transform these encoded values into actions.

## Deficits in the updating and representation of value are core computational deficits in Neurexin1α mutants

In order to reveal which aspects of the decision process were altered in Nrxn1α mutants, we took advantage of Q-learning models to quantitatively describe relevant drivers of choice in feedback-based reinforcement paradigms (*Daw, 2011*; *Sutton and Barto, 1998*). Our data suggest that choice abnormalities in Nrxn1α KO mice reflect deficits in the updating or encoding of choice values, encapsulated by reductions in the learning rate (α) and outcome differentiation (γ) parameters, as opposed to differences in how mice translate value into action (β) or persist on actions independent of outcome (κ) (*Figure 4*). These data are reminiscent of work from schizophrenic subjects in a probabilistic reinforcement learning paradigm, where similar modeling suggested a reduction in the learning rate in patients versus neurotypical controls (*Hernaus et al., 2018*). Of particular interest, these investigators interpreted alterations in learning rate not to reflect perturbations in the reward prediction error (RPE) signal itself but to changes in how those signals were integrated to update value for future actions (*Hernaus et al., 2019*; *Hernaus et al., 2018*). While we cannot directly map parameters of the reinforcement model to neural circuits, this interpretation suggests that relevant circuit loci might be those tasked with integrating dopaminergic RPE signals, including connections between cortical regions and the striatum.

## Deletion of Neurexin1α from telencephalic excitatory neurons recapitulates choice abnormalities of the constitutive knockout

The above hypothesis, together with robust expression of presynaptically expressed Nrxn1α throughout cortex and its known role in mediating excitatory synaptic function in hippocampal circuits, directed us toward probing its function in corticostriatal circuits. A large literature has implicated multiple excitatory forebrain populations in flexibly encoding the expected value of anticipated reward (*Kennerley and Walton, 2011*; *Rolls, 2000*; *Tremblay and Schultz, 1999*; *Wallis and Kennerley, 2010*; *Wallis and Miller, 2003*), reward-dependent modulation of working memory (*Wallis and Kennerley, 2010*), and forming associations between motivated behaviors and their outcomes (*Hayden and Platt, 2010*). Consistent with this, deletion of Nrxn1α from embryonic telencephalic excitatory neuron progenitors recapitulated the value-based deficits observed in the constitutive KOs (*Figure 5*). While we do not claim this as the sole circuit-specific deletion capable of generating this phenotype, some degree of specificity was demonstrated by the absence of decision-making phenotypes in our thalamic Nrxn1α deletion (*Figure 6*).

Unfortunately, the broad recombinase expression of the Nex-Cre transgenic within telencephalic excitatory populations precludes us from assessing the importance of Nrxn1α in specific telencephalic populations such as medial prefrontal or sensorimotor cortices. It also cannot rule out a role

for excitatory populations within the amygdala that have been linked to goal-directed instrumental actions (*Corbit et al., 2013*). Co-expression networks seeded by autism candidate genes have highlighted human mid-fetal deep layer cortical neurons from both prefrontal and primary motor/somatosensory cortices as potential sites of autism pathogenesis (*Willsey et al., 2013*). Furthermore, human patients with damage to the ventromedial prefrontal cortex exhibit similar deficits in value-based decision-making tasks as those seen in our Nex deletions (*Camille et al., 2011*; *Fellows and Farah, 2007*). Further assessment of the contribution of prefrontal Nrxn1α function to the observed phenotypes awaits Cre transgenic lines with both greater cortical regional specificity and embryonic expression. It is worth noting that Nex-Cre transgenic mice also label a small subset (~10%) of VTA neurons that project to the medial shell of the nucleus accumbens (*Bimpisidis et al., 2019*; *Kramer et al., 2018*). While we cannot formally rule out the contribution of these neurons to our behavioral results, they are unlikely to account for our Ca$^{2+}$ imaging results, as their projections are distant from our imaging site.

## Circuit-specific ablation of Neurexin1α disrupts value-modulated neural signals within striatum

Based on our behavioral data and computational modeling from multiple Nrxn1α mutants, expression patterns of Nrxn1α transcripts (*Fuccillo et al., 2015*), and the known pre-synaptic function of this molecule in maintaining synaptic connectivity (*Anderson et al., 2015*; *Aoto et al., 2013*; *Etherton et al., 2009*; *Missler et al., 2003*), we hypothesized that the observed value-based abnormalities resulted from altered synaptic transmission at key sites for integration of RPEs into action value coding. Putative circuit loci include: (1) connections within value-encoding forebrain excitatory areas; (2) connections from cortex onto mesencephalic dopamine neurons that encode striatal-targeting RPE signals (*Takahashi et al., 2011*); and (3) connections from cortical areas into striatum. Reasoning the aforementioned possibilities would all impact neural signals of striatal SPNs, we recorded population Ca$^{2+}$ activity of putative dSPNs via fiber photometry (*Figure 7A–C*). In support of this idea, we observed value-modulated signals leading up to trial initiation (*Figures 7D and F* and *8A,D, and H*), consistent with population Ca$^{2+}$ imaging signals observed in both SPN subtypes as mice approach palatable food (*London et al., 2018*). While our imaging does not provide the clarity of cellular-level approaches (*Donahue et al., 2019*; *Kwak and Jung, 2019*), it clearly resolved two phases of activity – a slow ramp occurring ~10 s before trial initiation and a fast peak in the 1 s leading up to initiation – that correlated with prior reward outcome and RPE (*Figures 7* and *8*). Interestingly, the Nex-Nrxn1α mutants displayed a clear disruption of these reward variable correlations with p-dSPN activity, specifically for the fast peak immediately preceding trial initiation (*Figure 8H*). We suggest a hypothesis wherein RPE signals are not appropriately integrated in Nex-Nrxn1α mutants, depriving striatal circuits of essential reward relevant information for subsequent action selection (*Hernaus et al., 2019*; *Hernaus et al., 2018*). Recent evidence from ex vivo brain slices suggests complex alterations to excitatory synaptic transmission for both anterior cortical and thalamic projections to striatum (*Davatolhagh and Fuccillo, 2020*). Nevertheless, in vivo neural recordings of both cortico-striatal and value-encoding cortical circuits during this task will be needed to understand how Nrxn1α mutations contribute to altered striatal representations of value.

Extensive associations have been found between mutations in Nrxn1α and a range of neuropsychiatric disorders (*Dachtler et al., 2015*; *Duong et al., 2012*; *Huang et al., 2017*; *Kirov et al., 2009*; *Rujescu et al., 2009*; *Sanders et al., 2015*; *Südhof, 2008*). Here we show that Nrxn1α plays a key functional role in specific forebrain excitatory projection circuits governing cognitive control of value-based action selection. It is interesting to speculate that the widespread nature of basic reinforcement learning abnormalities seen across neuropsychiatric diseases could be explained by similar network dysfunctions as seen here for Nrxn1α mutants. Further work will be necessary to test the generalizability of these observations for other neurodevelopmental psychiatric disorders and further refine the telencephalic excitatory populations of relevance.

# Materials and methods

## Contact for reagent and resource sharing

Code used for data analysis is available on the public Fuccillo lab github site (https://github.com/oalabi76/Nrxn_BehaviorAndAnalysis; *Alabi, 2020*; copy archived at swh:1:rev:b8233aab4e607f82c868caf2dfe4007790088e8e). Data for this manuscript is posted to Dryad (*Alabi et al., 2019*, Neurexin Photometry, Dryad, Dataset, https://doi.org/10.5061/dryad.vhhmgqnrq). Further information and requests for resources should be directed to and will be fulfilled by the Lead Contact, Marc Fuccillo (fuccillo@pennmedicine.upenn.edu).

## Experimental model and subject details

Animal procedures were approved by the University of Pennsylvania Harbor Laboratory Animal Care and Use Committee and carried out in accordance with National Institutes of Health standards. Constitutive Neurexin1α (Nrxn1α) KO mice were obtained from the Südhof lab (Stanford University) (*Geppert et al., 1998*). Nrxn1α$^{+/-}$ males and females were bred to produce subject for this study. In sum, 11 Nrxn1α$^{+/+}$ and 12 Nrxn1α$^{-/-}$ mice were used in this study. One Nrxn1α$^{-/-}$ mouse died in the early stages of training and its results were excluded. Nrxn1α conditional knockout mice were generated from sperm stock (Nrxn1 <tm1a(KOMP)Wtsi>) heterozygotes on the (C57Bl/6N background) obtained from the MRC Mary Lyon Center (Harwell, UK). The lacZ gene was removed via crosses to a germline-FLP recombinase, which was then bred off, followed by at least four generations breeding to homozygosity within our colony. Nex$^{Cre}$ mice (kind gift of Klaus-Armin Nave and Sandra Goebbels, Göttingen, Germany) were obtained and crossed onto Nrxn1α$^{c/c}$ mice (*Goebbels et al., 2006*). In this study 11 Nex$^{+/-}$ Nrxn1α$^{-/-}$, 14 Nex$^{-/-}$ Nrxn1α$^{c/c}$, and 13 Nex$^{+/-}$ Nrxn1α$^{c/c}$ mice were used. Olig3$^{Cre}$ mice were obtained (kind gift of Yasushi Nakagawa, University of Minnesota) and similarly crossed onto the Nrxn1α$^{c/c}$ colony (*Vue et al., 2009*). In this study 8 Olig$^{+/+}$; Nrxn1α$^{C/C}$ and 10 Olig$^{Cre/+}$; Nrxn1α$^{C/C}$ mice were used.

Whenever possible, animals were housed in cages with at least one littermate. One Neurexin1α wild-type and two Neurexin1α knockout animals were singly housed to avoid injury from fighting. Mice were food-restricted to maintain 85–90% of normal body weight and were given ad libitum access to water throughout the duration of the experiment. Mice were allotted 0.2–0.4 g of extra food on non-experimental days to account for the discrepancy in caloric intake from not receiving reward in a task. A 7 AM to 7 PM regular light–dark cycle was implemented for all mice used in this study. Cages were maintained in constant temperature and humidity conditions.

## Behavioral apparatus and structure

Experiments were conducted utilizing Bpod, a system specialized for precise measurements of mouse behavior (Sanworks LLC, Stony Brook, NY). A modular behavioral chamber (dimensions 7.5 L × 5.5 W × 5.13 H inches, ID: 1030) with three ports capable of providing light cues and delivering liquid rewards was used to measure behavioral events. Each port was 3D printed from clear XT Copolyester and housed an infrared emitter and phototransistor to measure port entries and exits precisely. Behavior chambers were enclosed in larger sound-attenuating boxes. For each behavioral paradigm, illumination of the center port after a 1 s intertrial interval indicated the beginning of a trial. Animals initiated trials by registering an entry to the lit center port, triggering a choice-period. The choice period was marked by the extinction of the center light and illumination of the ports on either side of the center. Mice were given an x-sec (varied by protocol) temporal window to enter either the left or right port and register a choice. Failure to register a choice in this period resulted in an omission, which was followed by a 3 s timeout and required the animal to reinitiate the task.

Successful registration of a choice resulted in the extinction of all port lights and the delivery of a variable volume of liquid supersac reward (3% glucose, 0.2% saccharin in filtered water) via a steel tube in the choice ports. Reward volumes and delivery probabilities were dependent on task conditions. The reward period lasted a minimum of 5 s. Following this mandatory minimum, the reward phase was extended if a mouse was noted to be occupying one of the three ports. The trial ended only after successful confirmation of port exit from all three ports. Reward volumes were regulated via individually calibrated solenoid valves, with specific time/volume curves to deliver precise reinforcement.

All port entries, exits, and other task events were recorded by the Bpod State Machine R1 (ID: 1027) and saved in MATLAB. Behavioral protocols and primary analysis were developed in MATLAB.

## Operant behavior

### Acquisition of goal-directed contingency

Mice were habituated to behavior chambers and ports over a 3-day period. Each day, animals were given a 10 min adjustment period followed by a program delivering 10 µL of reward every 30 s for 40 min. The first 40 trials were grouped into two blocks, with reward delivered either from the left or the right port for 20 contiguous trials. Following this period, reward was alternated between left and right port for the remaining 20 trials. Port lights were illuminated for a 10 s period to indicate reward delivery, followed by a 20 s ITI.

Following this introductory period, mice were introduced to a goal-directed task that required them to acquire a light-chasing reward contingency. Trials were initiated as described previously. During the choice phase, one of the two lateralized ports was illuminated at random. Mice were given 10 s to register a choice, or an omission was charged. If entries into the unlit lateral port or the center port were registered a 3 s timeout occurred and the animals had to reinitiate the trial until they selected the correct port. Successful selection of the correct port resulted in 10 µL of reward ($P_{rew}$ = 1.0). Sessions lasted 1 hr with no trial number limits. After 10 sessions, mice that had completed two consecutive days of >125 trials or 1 day >200 trials progressed to the serial reversal task. If mice missed this deadline, they were again assessed after their twelfth session. No mice failed to meet these criteria by the twelfth session.

### Serial reversal value task

After successfully acquiring the action–outcome contingency described above, mice progressed to a forced-choice two-alternative serial reversal paradigm with variable reward outcomes. Trial initiation occurred as described above, via entry into the central port. To ensure accurate initiation latencies, the state of the center port was assessed after the ITI. The beginning of a trial was delayed if a mouse was found occupying this port. Initiation of a trial led to a 5 s choice period in which both left and right lateral ports were illuminated as choice alternatives. Following selection, a variable volume of reward was delivered contingent upon current task conditions ($P_{rew}$ = 0.75 and 0.4 were used here). The reward phase lasted 5 s and trial termination did not occur till after mice successfully disengaged from all ports. One Nrxn1$\alpha^{-/-}$ mutant animal was excluded from the reversal study due to mis-calibrated solenoid valves.

Similar to our previous study, a 'moving window' of proximal task events was used to monitor mouse choice patterns (*Alabi et al., 2019*). Changes of choice-outcome contingencies were initiated when 8 of the last 10 actions were allocated to the large reward volume side. Following detection of this event, the lateralization of reward volumes was switched. These contingency reversals were un-cued and served to mitigate outcome-insensitive behavior. Reward probabilities were the same for both choices and consistent over a given session. The relative reward contrast was consistent over a given session. Eight reward environments were tested (four relative reward ratios across two reward reinforcement rates). Animals performed the eight tests in a random sequence, performing the high reinforcement sessions before the low reinforcement sessions. For initial introduction to task structure, mice were run in the reversal paradigm (12 µL vs. 0 µL) for 5–8 days prior to initiating the sequence of behaviors described above. All sessions were limited to 1 hr with no cap on trial number. Reward, however, was limited to 2000 µL in a session.

To ensure that behavioral measures were not overly influenced by spatial bias developed in one session (which could last for many subsequent sessions, across reward environments), sessions with excessive or carryover bias were excluded from this study and triggered a re-training phase before the experiment was continued. Bias was calculated as:

$$\mathrm{OverallBias} = (\mathrm{Pokes(Bias)} - \mathrm{Pokes(Non-Bias)})/\mathrm{Total\ Pokes}$$

where Pokes (Bias) denotes the number of port entries to the side which received more pokes and Pokes (Non-Bias) represents the number of pokes to the side that received less. A bias exceeding 0.45 initiated an automatic re-training phase lasting at least one session. Sessions with biases >0.2 triggered a watch-period in mice. If another session produced a bias >0.2 to the same spatial choice

alternative, that session was marked as having carry-over bias from a previous session and excluded – also triggering a retraining phase. Sessions were additionally excluded if animals met three conditions in a single session: (1) overall bias exceeding 0.45; (2) failure to complete a minimum of two contingency switches; and (3) failure to complete at least 100 selections of the nonbiased alternative. During re-training, animals performed one session of the 12 µL vs. 0 µL reversal task to eliminate spatial bias.

## Static contingency effort task

A behavioral paradigm with a stable reward contingency over 150 trials was used to assess how costs shape behavior. Cost was modeled as increased operant responding (FR3) before delivery of a reward. Costs were applied to one alternative for 150 trials, following which a relative reward reversal was initiated (10 µL vs. 0 µL) to eliminate the spatial bias developed during the task. Entry into one port during the choice phase led to extinction of the contralateral light. The chosen port remained lit until the animal completed the repetitive motor requirement necessary to obtain reward. Immediately upon completion of this requirement, reward was delivered as described previously. Equal reward volumes (8 µL, $P_{rew}$ = 1) were implemented during the experimental phase of this task. Trial structure was the same as in the reversal paradigm described above. All sessions were limited to 1 hr. Each animal performed two experimental sessions to account for potential spatial biases. One with the high motor threshold on the right and the other with it on the left choice port.

Before animals were exposed to relative costs, they were acclimated to the new behavioral requirements by a three-session minimum training period in which they completed this task with an FR3 vs. FR3 to increase response rate.

## Cognitive flexibility assays

To measure cognitive flexibility, we employed an attentional set shifting task where the correct port was first indicated by a lit visual cue and subsequently switched to a fixed egocentric spatial position. Trials were structured as previously described. In the first 25 trials, a light cue denoted the position of reward. Mice initiated trials in which one of the lateralized alternatives was illuminated, at random, during a 10 s choice window. Selection of the illuminated port resulted in a 10 µL reward, and selection of the unlit port resulted in a timeout. Following this baseline block, illumination of the choice ports continued to occur at random, but rewards were only delivered on one of the choice ports for the remainder of the session. Sessions were capped at 1 hr and 250 trials.

To further probe behavioral flexibility, we utilized an egocentric spatial reversal task. Individual trial structure was preserved. In the first block of 25 trials, one of the choice ports was assigned as the reward port. Following this introductory block, the opposite port was assigned as the reward port. On each trial, one of the two ports was illuminated at random. A 10 µL reward was given after selection of the appropriate port.

To account the potential biases and intersession fluctuations in performance, each animal was tested twice in each behavior – with alternating spatial cues in each session. $P_{rew}$ = 1 for both behaviors upon selection of correct alternative.

## Spontaneous alternation behavior

Mice were acclimatized to the testing room for 1 hr prior to testing. Alternating behavior was measured in a Y-maze (custom built, based on San Diego Instruments Y-maze 2005) and recorded with an overhead camera (10fps). To begin the test, each mouse was placed in arm C facing arms A and B. The mouse was allowed to freely explore the Y-maze for 5–8 min. If the mouse performed 15 arm entries (defined as entry of all four limbs into an arm) by the end of 5 min, the session was ended immediately. If the mouse had not performed 15 arm entries after 5 min, an additional 3 min was given. Mice that did not perform 15 arm entries within 8 min were excluded from the data. The video was manually scored by an experimenter who was blinded to the animal's genotype and sex.

## Analysis of behavioral performance

Data were analyzed using custom-written scripts developed in Matlab (*R Development Core Team, 2017*). We utilized basic function supplemented by the following toolboxes: Bioinformatics, Curve

Fitting, Data Acquisition, Global Optimization, Parallel Computing (*R Development Core Team, 2017*). Analytical code is available on request.

## Descriptive parameters

The session performance index was calculated as:

$$\text{Performance Index} = e \ln\left(\frac{\text{Pr(Large Reward)}}{1 - \text{Pr(Large Reward)}}\right)$$

where $Pr(\text{LargeReward})$ refers to the percentage of total choice that animals made to the large reward alternative over the course of a session.

The relative reward-stay of an outcome, A, versus another outcome, B, was calculated as:

$$\text{Relative Reward} - \text{Stay} = \ln\left(\left(\frac{\text{Pr(A)}}{1 - \text{Pr(A)}}\right) \middle/ \left(\frac{\text{Pr(B)}}{1 - \text{Pr(B)}}\right)\right)$$

where Pr(A) and Pr(B) refer to the probability that mice stay on the choice alternative producing outcome A and B, respectively, on the $t-1$ trial.

The adaptability index was calculated as:

$$\text{Adaptability Index} = \left(\sum_{i=1}^{n}\left(\left(L_i^{\text{post}} - S_i^{\text{post}}\right) + \left(L_i^{\text{pre}} - S_i^{\text{pre}}\right)\right)/10\right)/n$$

where $L_i^{\text{pre}}$ and $L_i^{\text{post}}$ refer to the number of large alternative selections in the 10 trials before and after the i-th contingency switch in an individual session and $S_i^{\text{pre}}$ and $S_i^{\text{post}}$ refer to the number of small alternative selections in the same time window. n is the number of blocks completed in a session.

The relative initiation latency was calculated as:

$$\text{Relative Latency to Initiate} = \left(\text{LatInit}_{\text{Large}} - \text{LatInit}_{\text{Small}}\right)/\text{LatInit}_{\text{Small}}$$

where $\text{LatInit}_{\text{Large}}$ and $\text{LatInit}_{\text{Small}}$ refer to the average latency to initiate trials following large reward and small reward outcomes, respectively, in an individual session.

## Logistic regression

We employed a logistic regression to model current choice as a function of past actions and outcomes (n = 5 trials):

$$\log\left(\frac{R(i)}{1-R(i)}\right) = \beta_0 + \sum_{p=1}^{n}\beta_p^{LR}LR(i-p) + \sum_{p=1}^{n}\beta_p^{SR}SR(i-p) + \sum_{p=1}^{n}\beta_p^{NR}NR(i-p) + $$
$$\sum_{p=1}^{n}\beta_p^{C}C(i-p) + \text{error}$$

where $R(i)$ is the probability of choosing the right-sided alternative on the $i$th trial. $LR(i-p)$, $SR(i-p)$, and $NR(i-p)$ refer to the outcomes of the $p$th trial before the $i$th trial. $LR(i-p)$ is defined such that $LR(i-p) = +1$ if an animal received a large reward resulting from a right press on the $p$th previous trial, $-1$ if an animal received a large reward resulting from a left press on the $p$th previous trial, and 0 if the animal did not receive a large reward on that trial. $SR(i-p)$ and $NR(i-p)$ are defined similarly for trials that resulted in small reward and no reward outcomes, respectively. $C(i-p)$ is an indicator variable representing the previous choice behavior of the mouse (C = 1 for right-sided choice and C = 0 for left-sided choice). These variables provide a complete accounting of the choice, reward history, and interaction of the two in our task. This method assumes equivalent reinforcement from outcomes regardless of the lateralization of choice. The model was fit to six random blocks of 85% of choice data. The coefficient produced by these blocks was averaged to produce individual coefficients for each animal. Regression coefficients were fit to individual mouse data using the *glmfit* function in Matlab with the binomial error distribution family. Coefficient values for individual mice were averaged to generate the plots shown in the supplemental figures.

## Reinforcement learning model

An adapted Q-Learning Reinforcement Model with five basic parameters was fit to the behavioral data produced by the relative reward serial reversal task (*Daw, 2011*; *Sutton and Barto, 1998*). Mouse choice patterns and outcome history were the primary inputs of the model. In order to capture trait-like characteristics of mouse behavior, behavioral sessions from the high and low reinforcement rate environments (four sessions each) were grouped and entered into the model together. The values of the lateralized choice alternatives were initiated at 0 and updated as follows:

$$Q_{t+1} = Q_t + \alpha(R_t - Q_t), \text{ where}$$

$$R_t = V_t^{\gamma}$$

In this model, $R_t$ is the value of the action taken on trial $t$ and $R_t$ is the function that approximates the perceived reward volume resulting from that action. $V_t$ is defined as a compressive transformation of the reward volume, $R_t - Q_t$, delivered after a choice raised to the coefficient, $\gamma$. $\gamma$ is the compression parameter that relates how sensitively mice respond to reward volumes of different magnitudes. $P_A(t) = \frac{1}{1+e^{-z}}$ , then, represents the reward prediction error (RPE) – the discrepancy between expected and realized reward – on trial $t$. The RPE is scaled by the learning rate ($\alpha$), which determines the extent to which new information about the state-action pairing alters subsequent behavior. The scaled RPE is then used to update the value of the chosen action for the subsequent trial $t+1$. The value of the unchosen alternative was not altered on any trial and did not decay.

We utilized a modified softmax decision function to relate calculated action values with choice probabilities. The probability of choosing an alternative A on trial $t$ was defined as:

$$P_A(t) = \frac{1}{1 + e^{-z}}, \text{ where}$$

$$z = \beta(Q_A(t) - Q_B(t)/12^{\gamma}) + \kappa C_{t-1} + c_{1-4}Env_{1-4}$$

The inverse temperature parameter, $\beta$, is the conversion factor linking theoretical option values with realized choice output. High values of $Q_A(t) - Q_B(t)$ indicate a tendency to exploit differences in action values, while lower values suggest more exploratory behavior. $12^{\gamma}$ is the value of alternative A relative to the value of alternative B. In order to compare $\beta$ across animals, this relative difference is scaled by $C_{t-1}$, representing the maximum Q value (as largest delivered reward was 12 µL). To account for the influence of proximal choice output on subsequent decisions, we included the parameter $\kappa$ – the persistence factor. This measure captures the extent to which the animal's choice on the $t-1$ trial influences its choice on the $t$ trial irrespective of outcome. $C_{t-1} = 1$ is an indicator variable that denotes whether the animal selected alternative A on the previous trial ($C_{t-1} = -1$) or if it selected alternative B ($\kappa$). To account for potential differences in bias between sessions, a bias term, $c_x$, with an indicator variable $Env_x$, was added for each session that the animal performed. This constant term captures spatial biases that animals have or develop in the course of a behavioral session. We performed a maximum likelihood fit using function minimization routines of the negative log likelihood of models comprised of different combinations of our three parameters ($\alpha$, $\beta$, $\gamma$, $\kappa$, c) in MATLAB (*Vo et al., 2014*). In order to resolve global minima, the model was initiated from 75 random initiation points in the parameter space.

## Fiber photometry

### Viral injection and fiberoptic cannula implantation

Trained Nex$^{+/-}$ Nrxn1$\alpha^{-/-}$ (n = 8) and Nex$^{+/-}$ Nrxn1$\alpha^{c/c}$ (n = 6) mice were injected with adeno-associated viruses and implanted with a custom fiberoptic cannula on a stereotaxic frame (Kopf Instruments, Model 1900). Anesthesia was induced with 3% isoflurane + oxygen at 1 L/min and maintained at 1.5–2% isoflurane + oxygen at 1 L/min. The body temperature of mice was maintained at a constant 30°C by a closed loop homeothermic system responsive to acute changes in internal temperature measured via rectal probe (Harvard Apparatus, #50–722F). After mice were secured to the stereotaxic frame, the skull was exposed and anatomical landmarks bregma and lambda were identified. The skulls of the mice were subsequently leveled (i.e. bregma and lambda in the same

horizontal plane) and 0.5 mm holes were drilled on regions of the skulls above the target locations. A pulled glass injection need was used to inject 300 nL of retroAAV2.EF1α−3xFLAG-Cre into the substantia nigra reticulata (SNr; AP: −4.2 mm, ML: +/−1.25 mm, DV: −3.11 mm) followed by 500 nL of AAV5.hSyn-DIO-GCamp6f into the dorsomedial striatum (DMS: AP: 0.85 mm, ML: +/−1.35 mm, DV: −2.85 mm). Holes were drilled ipsilaterally and injections were performed unilaterally per mouse. Virus was infused at 125 nL/min using a microinfusion pump (Harvard Apparatus, #70–3007) and injection needles were left in position for 10–20 min to allow diffusion of the viral bolus.

To implant each fiber optic, two 0.7 mm bore holes were drilled ~2 mm from the DMS skull hole. Two small screws were secured to the skull in these bore holes. A 400 μm fiberoptic cannula was lowered into the DMS injection site. Small abrasions on the skull surface were created with a scalpel, following which, we applied dental cement (Den-Mat, Geristore A and B) to secure the fiber optic placement. After surgery, mice were given oxygen at 2 L/min to aid in regaining consciousness. Mice were incubated for 4–6 weeks before recordings were performed. Approximately 2 weeks post-op mice were food deprived and reintroduced to the serial reversal task previously described. All data for photometry was collected only from 12 μL versus 0 μL sessions.

## Data acquisition

Before recording sessions, mice were attached to a fiber-optic patch cord (400 μm core, 0.48 NA; Doric Lenses) to enable recordings. Patch-cords were attached to a Doric 4-port minicube (FMC4, Doric Lenses) to regulate incoming and outgoing light from the brain. An LED light driver (Thor Labs, Model DC4104) delivered alternating blue (470 nm, GCamp6f excitation) and violet (405 nm, autofluorescence/movement artifact) light to the brain. Light was delivered at ~50μW. The resulting excitation emissions were transferred through a dichroic mirror, a 500–550 nm filter, and were ultimately detected by a femotwatt silicon photoreceiver (Newport, Model 2151).

After attachment to the fiber-optic, animals were given a 5 min window to recover from handling before the initiation of a session. All recorded mice were trained to perform the relative reward serial reversal task before surgery. Animals were reintroduced to the task ~2 weeks post-surgery. At 3 weeks, expression of the GCamp6f construct was assessed and animals were trained to perform the task with the attached fiber-optic. After a minimum of 4 weeks and three full training sessions with the fiber optic, animals were eligible for recordings. Sessions lasted 1 hr. We introduced a 0–1 temporal jitter after the ITI and before the choice period to aid in dissociating task events.

## Signal processing and analysis

Raw analog signals from behaving mice were demodulated (Tucker Davis Technologies, RZ5 processor) and recorded (Tucker Davis Technologies, Synapse). Demodulated 470 nm and 405 nm signals were processed and analyzed using custom Matlab (MathWorks, R2018b) scripts that are freely available upon request. Signal streams were passed through the *filtfilt* function, a zero-phase digital filter that filters data in both the forward and reverse direction to ensure zero phase distortion. Next, the data were down-sampled to 20 Hz. To account for bleaching of background autofluorescence in the patch cords over long recording sessions, the demodulated 470 nm and 405 nm signals were baselined to zero (the last value in the recording was used as an offset to have the signal decay to 0) and were fitted with cubic polynomial curves, which were subsequently subtracted from the signals. The ΔF/F of the debleached signals was calculated by sorting values into a histogram (100 bins) and then selecting the largest bin as the baseline signal. This baseline was subtracted from the raw 470 and 405 and then those values were divided by the baseline (note that the operation below was performed on both 470 and 405) [ΔF/F = (debleach(a)−baseline)/baseline]. Following this, the 405 nm control signal was subtracted from the 470 nm GCamp6f emission signal. The subtracted ΔF/F was transformed into z-scores by subtracting the mean and dividing by the standard deviation of a 2 min window centered on each point (1 min in front and behind). These standardized fluorescence signals were used for all subsequent analysis and visualization. The Bpod State Machine delivered electronic TTLs marking behavioral events to Synapse Software, which recorded their time and direction.

## Modeling signal dynamics

The dynamics of preinitiation signal components was modeled as function of action output in the form of upcoming choice behavior (choice lateralization relative to implant [*Choice*], stay/shift

behavior [*Stay*], explore/exploit behavior[*Explore*]), reward (reward volume on previous trial [*RewardHist*], reward prediction error [*RPE*] on previous trial and the relative action value on the current trial [$\Delta Q^*temp^{-1}$]), prior signal dynamics (the preinitiation slope and integral on the previous trial [*PIS* and *PIT*], respectively), and the latency to initiate trials [*LatInit*]. Because the slope occurs after the integral on every trial and because slope and integral components are anti-correlated, the preinitiation integral on the *t* trial was included as a regressor in the modeling of the slope component. To account for individual animal differences in preinitiation signal components, we utilized a linear mixed-model:

$$\mathrm{Preinitiation\,Integral} \sim \mathrm{Reward\,Hist} + \mathrm{RPE} + \Delta Q^*\mathrm{temp}^{-1} + \mathrm{Choice} + \mathrm{Stay} + \\ \mathrm{Explore} + \mathrm{PIT} + \mathrm{LatInit} + (1|\mathrm{Subject}) + 1$$

$$\mathrm{Preinitiation\,Slope} \sim \mathrm{Reward\,Hist} + \mathrm{RPE} + \Delta Q^*\mathrm{temp}^{-1} + \mathrm{Choice} + \mathrm{Stay} + \mathrm{Explore} + \\ \mathrm{PIS} + \mathrm{LatInit} + \mathrm{PreInitiation\,Integral} + (1|\mathrm{Subject}) + 1$$

## Histology and immunohistochemistry

Mice were perfused via the left ventricle of the heart with 10 mL of 90% formalin. Whole brains were isolated and post-fixed in formalin overnight; 50 µm coronal and sagittal slices were sectioned in PBS. Slices from mice included in behavioral experiments were immediately mounted on microscope slides for imaging on an automated fluorescence microscope (Olympus BX63) at 10× (Olympus, 0.4NA). Additional sections were blocked in 3% normal goat serum in PBS for 1 hr and incubated with primary antibody overnight (1:500 Chick anti-GFP, abcam 13970; 1:1000 Mouse anti-FLAG, Sigma F1804). The following day, slices were washed with PBS and incubated for 3 hr with secondary antibody (1:1000 Goat Alexa488-conjugated anti-Chick, abcam 150173; 1: 1000 Goat Alexa647-conjugated anti-Mouse, Invitrogen #A-21235). Slices were washed 3× with PBS for 30 min and mounted on slides. Images were acquired from the same epi-fluorescent microscope as other images.

## Electrophysiology

Mice were deeply anesthetized and perfused transcardially with ice-cold ACSF containing (in mM): 124 NaCl, 2.5 KCl, 1.2 NaH$_2$PO$_4$, 24 NaHCO$_3$, 5 HEPES, 12.5 glucose, 1.3 MgSO$_4$, 7H$_2$O, 2.5 CaCl$_2$. The brain was rapidly removed and coronal sections (250 µM thickness) were cut on a vibratome (VT1200s, Leica) in ice-cold ACSF. Sections were subsequently incubated <15 min in a NMDG-based recovery solution containing 92 NMDG, 2.5 KCl, 1.2 NaH$_2$PO$_4$, 30 NaHCO$_3$, 20 HEPES, 25 glucose, 5 sodium ascorbate, 2 thiourea, 3 sodium pyruvate, 10 MgSO$_4$, 7H$_2$O, 0.5 CaCl$_2$. The identity of retrogradely infected SPNs was visualized through viral fluorescence. Whole-cell recordings for mIPSCs were made using an internal solution containing (in mM): 135 CsCl, 10 HEPES, 0.6 EGTA, 2.5 MgCl, 10 Na-Phosphocreatine, 4 Na-ATP, 0.3 Na-GTP, 0.1 Spermine, 1 QX-314. mEPSCs were recorded using an internal solution containing (in mM): 115 CsMeSO$_3$, 20 CsCl, 10 HEPES, 0.6 EGTA, 2.5 MgCl, 10 Na-Phosphocreatine, 4 Na-ATP, 4 Na-GTP, 0.1 Spermine, 1 QX-314 (pH adjusted to 7.3-7.4 with CsOH). Miniature spontaneous events were recorded in the presence of Tetrodotoxin (TTX; 1 µM), 2,3-dioxo-6-nitro-1,2,3,4-tetrahydrobenzo[*f*]quinoxaline-7-sulfonamide (NBQX; 10 µM), D-(-)−2-amino-5-phosphonopentanoic acid (D-APV; 30 µM) for mIPSCs, and TTX plus picrotoxin (100 µM) for mEPSCs. Electrophysiology data was acquired using custom-built Recording Artist software (Rick Gerkin), Igor Pro6 (Wavemetrics), and analyzed using Minianalysis (Synaptosoft).

## Statistical methodology

Power analysis was conducted in G*Power 3.1.9.4 (*Faul et al., 2007*) to obtain the appropriate sample size for the comparison of relative reward stay values of Neurexin1α wild-type and mutant animals. A power analysis for repeated measures ANOVA with two groups (wild-type, mutant) and eight measurements (two reward probabilities, four relative reward ratios), at power of 0.80, an alpha level of 0.05, and a medium-large effect size ($f = 0.40$), indicated a required sample size of 12. The sample size, n, for each experiment is clearly labeled on figures and in figure legends. Animals were tested in a repeated design aimed to assess their reward sensitivity in various reward conditions. However, each reward condition was only recorded once per animal. Replicate information for

RNA experiments can be found in the methods section of the manuscript. Criteria for exclusion are detailed in the methods section as well.

All data were initially tested with appropriate repeated measure ANOVA (Prism8.0). Univariate regressions were performed in Prism8.0. Multivariate linear regressions were performed using the *fitlm* function in MATLAB. Multivariate linear mixed models were performed using the *fitlme* function in MATLAB. Main effect and interaction terms are described within figures, figure legends, and the results. Preinitiation slope coefficients were calculated using the *polyfit* function in MATLAB. The integral of photometry signals was calculated using the *trapz* function in MATLAB.

## Acknowledgements

This work was supported by grants from the NIMH (F31-MH114528 to OA, R00-MH099243 and R01-MH115030 to MVF), the Whitehall Foundation, and the IDDRC at the Children's Hospital of Philadelphia. We thank Boris Heifets and Elizabeth Steinberg for assistance with initial Matlab code for photometry analysis. We also thank Alexandria Cowell for excellent assistance in mouse colony genotyping. Finally, we thank Patrick Rothwell and members of the Fuccillo lab for comments on the manuscript.

## Additional information

### Funding

| Funder | Grant reference number | Author |
| --- | --- | --- |
| National Institutes of Health | R00MH099243 | Marc Vincent Fuccillo |
| National Institutes of Health | R01MH115030 | Marc Vincent Fuccillo |
| National Institutes of Health | F31MH114528 | Opeyemi O Alabi |
| Children's Hospital of Philadelphia | IDDRC Young Investigator | Marc Vincent Fuccillo |

The funders had no role in study design, data collection and interpretation, or the decision to submit the work for publication.

### Author contributions

Opeyemi O Alabi, Conceptualization, Data curation, Software, Formal analysis, Funding acquisition, Investigation, Visualization, Methodology, Writing - original draft, Writing - review and editing; M Felicia Davatolhagh, Luigim Vargas Cifuentes, Data curation, Formal analysis, Investigation; Mara Robinson, Michael P Fortunato, Data curation, Investigation; Joseph W Kable, Software, Formal analysis, Supervision, Writing - review and editing; Marc Vincent Fuccillo, Conceptualization, Supervision, Funding acquisition, Visualization, Methodology, Writing - original draft, Project administration, Writing - review and editing

### Author ORCIDs

Opeyemi O Alabi https://orcid.org/0000-0001-6732-2897
M Felicia Davatolhagh https://orcid.org/0000-0002-0607-5164
Luigim Vargas Cifuentes https://orcid.org/0000-0002-0831-0543
Marc Vincent Fuccillo https://orcid.org/0000-0002-6569-706X

### Ethics

Animal experimentation: This study was performed in strict accordance with the recommendations in the Guide for the Care and Use of Laboratory Animals of the National Institutes of Health. All of the animals were handled according to approved institutional animal care and use committee (IACUC) protocols (#805643) of the University of Pennsylvania.

Decision letter and Author response
Decision letter https://doi.org/10.7554/eLife.54838.sa1
Author response https://doi.org/10.7554/eLife.54838.sa2

## Additional files

### Supplementary files

- Source data 1. Table of Summary Statistics.

- Transparent reporting form

### Data availability

Source files have been placed on Dryad (Alabi, Opeyemi (2020), Neurexin Photometry, Dryad, Dataset, https://doi.org/10.5061/dryad.vhhmgqnrq) and code is at Fuccillo lab Github account (https://github.com/oalabi76/Nrxn_BehaviorAndAnalysis; copy archived at https://archive.softwareheritage.org/swh:1:rev:b8233aab4e607f82c868caf2dfe4007790088e8e/).

The following dataset was generated:

| Author(s) | Year | Dataset title | Dataset URL | Database and Identifier |
| --- | --- | --- | --- | --- |
| Alabi OO | 2020 | Neurexin Photometry | https://doi.org/10.5061/dryad.vhhmgqnrq | Dryad Digital Repository, 10.5061/dryad.vhhmgqnrq |

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
