## [Decision Letter]

**Acceptance summary:**

Behaviors that are oriented toward obtaining a specific goal are essential for normal life decision making but are impaired in neuropsychiatric disorders but the precise molecules that contribute to disrupted goal directed behaviors are unclear. This study demonstrates that removing Neurexin 1, a molecule that acts at the synapse to connect neurons, in excitatory brain circuits can disrupt goal directed behaviors and neural signals relating to reinforcement of these behaviors. These studies provide new information into the molecules that may underlie disrupted decision making processes in neuropsychiatric disorders.

**Decision letter after peer review:**

Thank you for submitting your article "Disruption of Nrxn1a within excitatory forebrain circuits drives value-based dysfunction" for consideration by *eLife*. Your article has been reviewed by three peer reviewers, and the evaluation has been overseen by a Reviewing Editor and Michael Frank as the Senior Editor. The following individual involved in review of your submission has agreed to reveal their identity: Talia Lerner (Reviewer #2).

The reviewers have discussed the reviews with one another and the Reviewing Editor has drafted this decision to help you prepare a revised submission.

Summary:

This study examines the role of Neurexin1a, a neuropsychiatric disorder-associated protein, in value-based decision making in mice. The authors address this by using a Neurexin 1a knockout and demonstrate deficits in optimizing selection strategies in an operant task with variable reward outcomes in this mouse line. Similar behavioral deficits are observed with a Neurexin1a conditional knockout (cKO) crossed to the Nex1-Cre, a strain that is supposed to express Cre-recombinase in postmitotic progenitors of forebrain projection neurons. Further, using fiber photometry paired with genetically encoded calcium sensors the authors show that Nex-Cre-Neurexin1a-cKO mice display deficits in activity of striatal direct pathway spiny projection neurons (dSPNs) prior to choice selection. Overall this study characterizes a genetic contribution to circuit dysfunction for neuropsychiatric disease relevant behaviors, pertaining to how animals choose actions according to cost and benefit.

Essential revisions:

1) There are concerns with the specificity of the genetic approach used. It appears that the conditional genetic approach was used to allow for developmental knockout of Nrxn1a as opposed to deletion in the adult animal. Can the author's please clarify and justify the need for using the developmental knockout approach? Further, addressing the developmental role of Nrxn1a and the human literature if applicable could improve the interpretations of the genetic approach used. Additionally, the Nex-Cre mice that are used for deletion of Nrxn1a from forebrain projections, has also been reported in mid- and hindbrain structures. Thus, it is difficult to interpret if behavioral abnormalities in these mice are due to deletion of Nrnx1a from forebrain projections or if they arise due to deletion of Nrxn1 in other regions of the brain. There is also misleading use of "cortical" rather than "forebrain" in the manuscript. Overall, clarification in text edits would improve the rationale for this genetic approach and clarify the specificity of the Nex-Cre approach.

2) There are concerns with the fiber photometry data that should be addressed. The z score values are low. In peak detection algorithms typically thresholds for something being defined as a significant peak at least 2.0 standard deviations above the median or mean average deviation of the data (some studies even use a cutoff of 2.9). However, here the Zscores are 0.2. It is difficult to determine if this is due to how Zscores were calculated or if there are not actually significant events detected. To address this please plot control data – either with a fluorophore (eYFP) or an isosbestic control (although eYFP is preferable) to ensure that fluctuations are due to detected calcium and not just pH shifts or movement artifacts that are occurring during the task.

Additional concerns are the difference in photometry signals between Nex-control and Nex-cKO animals in Figure 8, which could be due to the additional injection of cre (retroAAV2.EF1α-3xFLAG-Cre) in Nex-Nrxn1a cKO animals. This could potentially affect neurexin expression and activity of the direct pathway SPNs. Another control (Nrxnc/c; Nex+/+) would help to resolve this concern.

3) There is misleading use of "reward-associated neural signals" and "value-related neural signals" in reference to dSPN activity prior to the initiation of a trial. While the activity is interesting it does not relate to reward presentation or consumption. Please also show dSPN activity at the time of choice and reward consumption in addition to trial initiation, as the activity at time of choice relates to decision making and activity at the time of reward relates to value-encoding.

4) There are concerns that the Neurexin1a-KO mice have a working memory deficit. The t-1 regression coefficient is lower and other coefficients aren't higher, suggesting that the KOs are not using as much information from past trials to guide their actions as controls are. Furthermore, the failure to modulate initiation latency based on previous outcome could be because they don't remember it easily. Finally, the differences between the control and KO mice are more apparent with lower probability of reward, perhaps because it's harder for them to remember with so much uncertainty. Two experiments could help control for this: (1) test working memory directly e.g. in a spontaneous alternation task, and (2) run a version of the authors' task with P=1. When P=1, uncertainty is not an issue, but the representation of two choices and their values is.

5) There are concerns with the interpretation of deficits in responses to relative reward outcomes in the Neurexin1a-KO. In Figure 1, Neurexin1a-KO animals show deficits in responses to relative reward outcomes. Specifically, the largest deficit observed was at the peak of the deltaSucrose. One interpretation is that Neurexin1a-KO animals simply consume less sucrose than WT's and therefore are less motivated to select the higher sucrose volumes, not because there is a deficit in optimizing operant strategies. Thus additional experiments examining whether Neurexin1a-KO mice consume comparable amounts of sucrose as WT animals in a standard operant paradigm or in a free-access model would help to address this concern.

6) There is a disconnect between the behavioral results in the Nex-Neurexin1a-cKO animals (which the authors state are targeted to forebrain neurons) and subsequent neural imaging studies targeted to direct pathway neurons of the dorsal striatum. While the authors discuss the importance of the dorsal striatum in reward-based tasks, they fail to address how Neurexin1a deficits in the prefrontal cortex/forebrain would lead to deficits in dorsal striatum activity during the task. Please provide clarification in the text.

7) Many of the findings remain largely correlative in nature. While the reviewers do not see a need for further experimentation to show a gain of function the authors should carefully examine some conclusions to indicate the correlative but non-causal results.

8) Some of the figures should be edited for clarity. For example, in some figures such as 1B, the statistical differences are entirely unclear. Although there is a described overall genotype difference, it is unclear if there was an interaction and at what reward size these differences were statistically important. This lack of clarity can be seen in most of the figures. Further, based on the information presented here, it is not possible to determine interactions or how post-hoc analysis was completed. While the figure legends contain some statistical information, they do not lead the reader to fully understand the important changes. Please also include how any post-hoc comparisons were done.

[Editors' note: further revisions were suggested prior to acceptance, as described below.]

Thank you for submitting your article "Disruption of Nrxn1a within specific excitatory forebrain circuits drives value-based dysfunction" for consideration by *eLife*. Your article has been reviewed by three peer reviewers, and the evaluation has been overseen by a Reviewing Editor and Michael Frank as the Senior Editor. The following individual involved in review of your submission has agreed to reveal their identity: David Dietz (Reviewer #3).

The reviewers have discussed the reviews with one another and the Reviewing Editor has drafted this decision to help you prepare a revised submission.

Summary:

This study examines the role of Neurexin1a, a neuropsychiatric disorder-associated protein, in value-based decision making in mice. The authors address this by using a Neurexin 1a knockout and demonstrate deficits in optimizing selection strategies in an operant task with variable reward outcomes in this mouse line. Similar behavioral deficits are observed with a Neurexin1a conditional knockout (cKO) crossed to the Nex1-Cre, a strain that is supposed to express Cre-recombinase in postmitotic progenitors of forebrain projection neurons. Further, using fiber photometry paired with genetically encoded calcium sensors the authors show that Nex-Cre-Neurexin1a-cKO mice display deficits in activity of striatal direct pathway spiny projection neurons (dSPNs) prior to choice selection. Overall this study characterizes a genetic contribution to circuit dysfunction for neuropsychiatric disease relevant behaviors, pertaining to how animals choose actions according to cost and benefit.

Essential revisions:

While the revisions have improved the study there are additional revisions needed. Below the remaining concerns are listed including how each should be addressed by tempering the language, re-analysis of data, or the inclusion of one new additional experiment.

1) Please address the below comment and comment #2 by performing re-analysis with the collected data and providing additional details. New experiments are not required.

Even with the additional control data provided, the fiber photometry data, as presented, is difficult to interpret. The Z-score peaks presented in Figure 7D-F (0.2 Z scores) are significantly lower than peaks seen in the overall representative trace in Figure 7C (where there are peaks of up to ~4 Z-scores). While the authors suggest that the small z-scores are due to genetically defined populations, work from several other groups recording from comparable cell populations have seen much larger task-specific peaks in GCaMP-measured activity (Cui et al., 2013 Nature; Cui et al., 2014 Nat. Prot.; Klaus et al., 2017 Neuron; Calipari et al., 2016 PNAS), demonstrating that this alone cannot explain this. There are several levels of data analysis that are missing from the manuscript that would alleviate concerns regarding this issue:

a) Changes in Ca^2+^ signaling should be presented in a trial-by-trial basis in addition to the summary seen in Figure 7.

b) Trial-specific traces should be presented in a non-Z-scored fashion (see comment 2 for why this is particularly important).

c) Although the data from the 405 control channel is presented, it is presented as an internally controlled Z-score. The 405 traces should be plotted at the same scale as the 470 trace to preclude the signal detected being movement artifacts. There is no mention of scaling in the analysis pipeline and this is critical to using the 405 as a control.

d) The authors report the laser power at 50μW, which is very low for both recording channels – especially the 405 – and could in part explain the flat line for the 405 channel. i.e. the light power was not high enough to detect events in that channel.

e) In the Materials and methods the authors mention that the data is filtered but provide no additional information about the filter. What is the filter, what is the equation, how does this filter alter the timing of the signal relative to the events? This is critical to understand the analysis pipeline.

2) The biggest issue in the data processing is with how the z-scores were calculated for the peri-event records. Because the signals were done as a change from the entire peri-event trace this makes it so that the median of the trace is 0. For signals with bigger peri-event records this will just move the pre and post response traces down. It will look like the peaks are similar, but the ramp up and down are different. This would then make the data look like the complete opposite of the raw data. The reason this is concerning here is that they data for the largest event shows this pattern, where the baseline is lower, rather than the event. Nearly all of the work on striatal populations has shown that these circuits scale with reward value – with larger peak response around the larger outcome. Thus, these data may actually show the opposite findings when plotted in a more appropriate way.

3) Please address the below concern by tempering the language in the manuscript.

Although the authors address concerns regarding specificity of the NEX-Cre/Nrx1aKO mouse line and potential mechanistic link between forebrain function in the text, there remains a large disconnect between the widespread loss of neurexin1a expression and the specific effects seen in the DMS->SNr circuitry. As the authors mention, the NEX-Cre has primary recombination sites in the amygdala, hippocampus, and several PFC subregions; each which feed into the DMS-> SNr pathway and each provide a unique aspect to reward-associated behavior. A key premise of the manuscript is that loss of forebrain neurexin1a leads to disruption of the DMS->SNr circuitry during reward-seeking. However, what is missing from the manuscript is how is this circuit disruption occurring. Given the widespread loss of neurexin1a expression, it is difficult to infer a potential mechanism or pathway as a source for DMS activity disruption, beyond likely glutamatergic input.

4) Please address the below concerns by performing an additional experiment to examine mEPSCs.

A main lingering concern is still about the retroAAV cre injection into SNr. The mIPSC recordings in DMS show that one particular change (in inhibitory transmission) does not occur but doesn't rule out other possibilities, particularly in terms of potential changes in excitatory transmission. It is assumed that the mEPSCs in Nex-cre Nrxn1a-cKO mice change in comparison to Nex-cre controls (that is their hypothesis, that corticostriatal deficits occur and underlie the changes in dSPN activity), although they don't show it but reference unpublished data in the reviewer response. A retroAAV cre injection in Nrxn1a cKO-only animals (no Nex-cre) to determine that mEPSCs don't change if the Nex-cre isn't there would be important. This would show that retroAAV cre expression doesn't do anything additional that is not intended at their synapse of interest.

More generally, injection of the retroAAV cre virus into Nrxn1a-cKO mice in SNr would cause recombination in SNr as well as in all input structures of SNr (not only DMS). The basal ganglia are so interconnected this could cause strange unforeseen consequences for dSPN activity.

[Editors' note: further revisions were suggested prior to acceptance, as described below.]

Thank you for submitting your article "Disruption of Nrxn1a within excitatory forebrain circuits drives value-based dysfunction" for consideration by *eLife*. Your article has been reviewed by three peer reviewers, and the evaluation has been overseen by a Reviewing Editor and Michael Frank as the Senior Editor. The following individual involved in review of your submission has agreed to reveal their identity: David Dietz (Reviewer #3).

The reviewers have discussed the reviews with one another and the Reviewing Editor has drafted this decision to help you prepare a revised submission.

Summary of the work:

Behaviors that are oriented toward obtaining a specific goal are essential for normal life decision making but are impaired in neuropsychiatric disorders. The precise molecules that contribute to disrupted goal directed behaviors are unclear. This study demonstrates that removing Neurexin 1, a molecule that acts at the synapse to connect neurons, in excitatory brain circuits can disrupt goal directed behaviors and neural signals relating to reinforcement of these behaviors. These studies provide new information into the molecules that may underlie disrupted decision making processes in neuropsychiatric disorders.

Recommendations:

It is suggested that the data presented in the response to reviewers be accessible so that the readers can see the raw data minimally processed. A concern hinges upon data that is included in the response where the z-scored data is introducing changes into the data that is not seen in the raw df/f data. Author response image 5 clearly shows that there are significant differences between A – unprocessed data and B- z-scored data that makes it look like huge decreases are happening that do not seem apparent in the unprocessed data. While this could be fine if the authors are transparent with their analysis pipeline, providing the data would help the reader draw their own conclusions.

It is recommended that the authors include this data or make it accessible to the readers in some form (i.e. link, website, etc).

---

## [Author Response]

Essential revisions:1) There are concerns with the specificity of the genetic approach used. It appears that the conditional genetic approach was used to allow for developmental knockout of Nrxn1a as opposed to deletion in the adult animal. Can the author's please clarify and justify the need for using the developmental knockout approach? Further, addressing the developmental role of Nrxn1a and the human literature if applicable could improve the interpretations of the genetic approach used.

Given Nrxn1a’s early expression around the onset of synaptogenesis, its role in initial synaptic maintenance and our desire to specifically relate our behaviors to the constitutive LOF allele, we chose gene removal early in development. While we agree that testing the continued necessity of Nrxn1a in adult structures is an interesting question, we believe it goes beyond our initial question of modeling Nrxn1a-associated brain dysfunction, which involves gene loss from early post-neurogenic periods. These specific adult maintenance functions will be pursued in a separate study. We have added text discussing the early expression of Nrxns prior to synaptogenesis and our desire to test this early function given its relevance to disease models.

Additionally, the Nex-Cre mice that are used for deletion of Nrxn1a from forebrain projections, has also been reported in mid- and hindbrain structures. Thus, it is difficult to interpret if behavioral abnormalities in these mice are due to deletion of Nrnx1a from forebrain projections or if they arise due to deletion of Nrxn1 in other regions of the brain. There is also misleading use of "cortical" rather than "forebrain" in the manuscript. Overall, clarification in text edits would improve the rationale for this genetic approach and clarify the specificity of the Nex-Cre approach.

Regarding the specificity of the Nex-Cre allele, our own in-lab analysis together with analysis of the original paper (Goebbels et al., 2006) suggest that expression in mid and hindbrain structures is minimal. While Table 1 in the original publication lists an enormous number of sites of recombination, it makes no distinction between sparse recombination (observed in mid/hindbrain, see Figure 3, Goebbels et al., 2006) and the primary recombination sites (cortex, hippocampus and amygdala). Recent publications (Kramer et al., 2018 and Bimpisidis et al., 2018) have highlighted a small (~10%) subpopulation of mVTA neurons that are NEX positive and captured by the Nex-Cre allele. While we cannot formally rule out a contribution of these neurons to our observed behavior, they are unlikely to contribute to our Ca^2+^ imaging findings, as this NEX-Cre DA subpopulation has been shown to project exclusively to the medial shell of the NAc (see Figure 3B, Kramer et al., 2018). Nex-negative DA neurons make up the entire population projecting to the DMS, where all of our recordings were performed. We have added a discussion of these caveats in our Discussion section.

Finally, we agree that our terminology needs to be clarified for publication. Beyond the confusion created by our interchanging “cortical” and “forebrain,” we realized that “forebrain excitatory neurons” describes both cortical, hippocampal, amygdala AND thalamic projections. We have decided on using “telencephalic excitatory” as this includes the neocortical, hippocampal and subset of amygdalar neurons captured by the Nex-Cre allele.

2) There are concerns with the fiber photometry data that should be addressed. The z score values are low. In peak detection algorithms typically thresholds for something being defined as a significant peak at least 2.0 standard deviations above the median or mean average deviation of the data (some studies even use a cutoff of 2.9). However, here the Zscores are 0.2. It is difficult to determine if this is due to how Zscores were calculated or if there are not actually significant events detected. To address this please plot control data – either with a fluorophore (eYFP) or an isosbestic control (although eYFP is preferable) to ensure that fluctuations are due to detected calcium and not just pH shifts or movement artifacts that are occurring during the task.

Z scores were calculated as the signal-mean/standard deviation of the signal over a moving 60 sec. window, to minimize the effect of broader signal fluctuations. The signals shown in Figure 7/8 and those used for correlation analysis do not employ peak detection algorithms, but instead peri-event averaged signals. The small signal fluctuations seen here likely result from the sparseness of the retrograde labeling employed to distinguish dSPNs and the relatively small motor movements required in our BPod behavioral machines, where mice can register choices with minimal motor output. Here we provide data on our 405 isosbestic channel demonstrating the absence of similar waveforms on this Ca^2+^ independent channel, highlighting the specificity of our 470nm signal. We have added this analysis in Figure 7—figure supplement 1A which accompanies main Figure 7, and a reference in the text.

Furthermore, we include in Author response image 1 examples of Ca^2+^ waveforms acquired in Med Associates boxes with a non-retrograde approach to label dSPNs (direct DIO-GCamp6f into D1Cre transgenic). While these averaged waveforms exhibit larger z-scored signals (note changed y-axis), they have very similar waveform shapes in the period leading up to initiation (compare gray shaded boxes, which bracket the same time window around task initiation).

**Author response image 1. sa2fig1:** 

Finally, we opted to analyze total signal waveforms as opposed to peaks because (1) it involved less transformation of the original photometry data; (2) the unclear significance of event-associated peaks; (3) our stringent criteria for peak detection lead to the omission of many trials in which initiation-relevant peaks were undetected. Nevertheless, to address reviewer concerns, we have performed a similar analysis using an event-associated peak detection algorithm focusing on a 1sec. window prior to initiation. To check our key results with this alternative analysis, we examined event-associated Ca^2+^ peak amplitude correlations with trial-by-trial behavioral parameters as in Figure 8. Consistent with our prior analysis, we found that multiple behavioral variables of value processing exhibit significant correlations with pre-initiation event-detected peaks and that these correlations are absent in Nex cKOs (see Author response image 2). Given the smaller sampling of data from event-associated peak-detection analyses and greater confidence in our waveform data, we opted to omit these data from the final manuscript. Nevertheless, the main take-home point that value-correlated neural signals are abrogated in Nex cKOs stands regardless of analysis method used.

Additional concerns are the difference in photometry signals between Nex-control and Nex-cKO animals in Figure 8, which could be due to the additional injection of cre (retroAAV2.EF1α-3xFLAG-Cre) in Nex-Nrxn1a cKO animals. This could potentially affect neurexin expression and activity of the direct pathway SPNs. Another control (Nrxnc/c; Nex+/+) would help to resolve this concern.

We acknowledge this potential caveat which resulted from a desire to specifically label one subtype of striatal SPN, the existing tools (no robust retroAAV2-FLP virus), and concern at the time for behavioral phenotypes in the Nex-Cre line, which had never been used in these types of behaviors before. Nevertheless, we do not believe that this design is the likely cause of our results for the following reasons: (1) the differing timelines for gene removal between the Nex-mediated recombination (early embryonic) as compared to rAAV2-Cre (2-3 weeks before behavior); (2) Nrxn1a expression levels in striatal SPNs are several orders of magnitude lower than that in cortex (Fuccillo et al., 2015) making LOF effects less likely; (3) given Nrxn’s exclusive presynaptic function, the fact that retrograde Cre will cause deletion in the recorded SPNs is unlikely to alter excitatory synaptic transmission that largely drives recorded Ca^2+^ signals in this population; (4) while the sparse nature of the retrograde labeling is unlikely to impact a large enough number of dSPNs to significantly alter their lateral inhibition, we directly tested this possibility by injecting rAAV2-3xFlag-Cre or rAAV2-EGFP-∆Cre (an enzymatically inactive truncated version of Cre) into Nrxn1a^C/C^; Nex^Cre^ adults, waiting the same duration of time as for our behavior/recording analysis (2-3 weeks) and then recorded mIPSCs in acute slices cut from these mice. We did not note any significant change in the frequency or amplitude of mIPSCs in neurons with Cre or δ-Cre expression, suggesting alterations in inhibition are unlikely to account for our changes in recorded Ca^2+^ signal. Taken together, we believe the most straight-forward explanation of our data is that deletion of Nrx1a from cortical progenitors alters the excitatory synapses of these projections pathways, with one result being the abrogation of value-related signals in a key target, the DMS. We have included this data in a new Figure 8—figure supplement1 (accompanies Figure 8) Furthermore, we have added a discussion of this experimental caveat to our revised Results.

3) There is misleading use of "reward-associated neural signals" and "value-related neural signals" in reference to dSPN activity prior to the initiation of a trial. While the activity is interesting it does not relate to reward presentation or consumption. Please also show dSPN activity at the time of choice and reward consumption in addition to trial initiation, as the activity at time of choice relates to decision making and activity at the time of reward relates to value-encoding.

We agree that the activity presented does not relate to reward presentation or consumption, as it is leading up to the initiation of the trial. We focused on this epoch for several reasons: (1) it is the most clearly temporally dissociated signal, as compared with the choice and reward-retrieval signals which have shorter and less variable inter-event latencies (200-400msec), making them challenging to parse using GCamp-based Ca^2+^ indicators; (2) we have multiple lines of unpublished evidence (in two different behavioral settings) that these pre-initiation signals are strongly modulated by the upcoming/current value of trials. While we agree that choice-associated signals might also be relevant, we do not think this invalidates our decision to focus on the initiation period, as this is the part of the behavior that demonstrates the greatest variability according to prior outcomes, with the choice and reward retrieval being more sterotyped actions resulting from extensive training in this specific paradigm. For revision, we have edited out the use of “reward-associated neural signals,” instead using “value-modulated neural activity,” or “value-modulated initiation activity,” as appropriate. Furthermore, as requested, we have provided the choice/retrieval-associated activity as requested in Figure 7—figure supplement 1B-E. While the downward choice waveforms are completely separated in control and overlapping in Nex cKOs, these differences likely relate to the preceding signal differences in value-modulated initiation activity. To quantify the choice-related signal we fit the choice aligned waveforms with a linear model and compared the slope values (Figure 7—figure supplement 1D) for control and Nex cKO mice (Figure 7—figure supplement 1E). We did not observe modulation of this slope by prior outcome or across genotype, suggesting that p-dSPN population Ca^2+^ activity at choice is not strongly modulated by value.

4) There are concerns that the Neurexin1a-KO mice have a working memory deficit. The t-1 regression coefficient is lower and other coefficients aren't higher, suggesting that the KOs are not using as much information from past trials to guide their actions as controls are. Furthermore, the failure to modulate initiation latency based on previous outcome could be because they don't remember it easily. Finally, the differences between the control and KO mice are more apparent with lower probability of reward, perhaps because it's harder for them to remember with so much uncertainty. Two experiments could help control for this: (1) test working memory directly e.g. in a spontaneous alternation task, and (2) run a version of the authors' task with P=1. When P=1, uncertainty is not an issue, but the representation of two choices and their values is.

It is true that a deficit in working memory (specifically relevant here as to which choice was previously made) can generate phenotypes similar to abnormalities in value coding for a task in which choice is the only read-out of internal processing we have. However, we don’t believe our data support this conclusion for the following reasons: (1) the lower coefficient for the large reward from the logistic regression models reflects the diminished reinforcing property of the reward; (2) if the mice had working memory deficits, we would also expect differences in the choice regression coefficient, which are not observed; (3) if the deficits were driven by working memory deficits, we would expect uniform choice abnormalities across all reward contrasts, instead we observe trends in which the largest choice differences by genotype are for the largest differences in reward contrast. However, to more directly address the reviewer’s concern, we have performed spontaneous alternation as requested on our Nex cKOs and their littermate controls. Consistent with normal working memory function in Nex cKOs, we do not observe any genotypic differences. These data have now been included in modified Figure 5—figure supplement 1G and in the Results.

5) There are concerns with the interpretation of deficits in responses to relative reward outcomes in the Neurexin1a-KO. In Figure 1, Neurexin1a-KO animals show deficits in responses to relative reward outcomes. Specifically, the largest deficit observed was at the peak of the deltaSucrose. One interpretation is that Neurexin1a-KO animals simply consume less sucrose than WT's and therefore are less motivated to select the higher sucrose volumes, not because there is a deficit in optimizing operant strategies. Thus additional experiments examining whether Neurexin1a-KO mice consume comparable amounts of sucrose as WT animals in a standard operant paradigm or in a free-access model would help to address this concern.

While it is true that the largest choice deficits noted are at the peak of the reward contrast (see point #4 above), we do not believe this has to do with the Nrx1a mutants consuming less sucrose. As we show in additions to Figure 2—figure supplement 1, reward probability and reward contrast, but not genotype determine the total volume of reward consumed per session (Figure 2—figure supplement 1B). In addition, the latency to initiate trials over time (for pr75, 12v0) also suggests that knockout mice do not lose engagement with the task as it progresses (Figure 2—figure supplement 1C). We have added reference to these data.

6) There is a disconnect between the behavioral results in the Nex-Neurexin1a-cKO animals (which the authors state are targeted to forebrain neurons) and subsequent neural imaging studies targeted to direct pathway neurons of the dorsal striatum. While the authors discuss the importance of the dorsal striatum in reward-based tasks, they fail to address how Neurexin1a deficits in the prefrontal cortex/forebrain would lead to deficits in dorsal striatum activity during the task. Please provide clarification in the text.

Our thinking here was based on three key facts: (1) Neurexin1a is highly expressed in deep-layer cortical neurons that send projections to striatum; (2) In hippocampal synapses, Neurexin1a KOs exhibit impaired excitatory synaptic transmission; (3) unpublished data from the lab demonstrating excitatory synaptic deficits in corticostriatal circuits of Neurexin1a KO and heterozygote mutant mice. We have added this clarification to the text, both in the Results section and in the Discussion.

7) Many of the findings remain largely correlative in nature. While the reviewers do not see a need for further experimentation to show a gain of function the authors should carefully examine some conclusions to indicate the correlative but non-causal results.

We do not feel that the initial characterization of the constitutive KO mice (Figure 1-4), nor the cell type-specific perturbations (Figure 5,6) are in any manner correlative. In fact, circuit-specific genetic manipulations are rarely, if ever, analyzed in such complex behavioral tasks. We do acknowledge that the analysis of neural signals in the Nex-specific deletions relies heavily on associations between population Ca^2+^ signals and aspects of behavioral performance (Figure 7,8). Given the novelty of our discovery, we felt it was important to uncover potentially relevant sites of Nrx1a-altered value coding before exploring neural circuit manipulations to prove causality. These complex experiments are currently underway in follow-up work to this study. For now, the best we can do is clarify the distinctions between our causal and correlative data throughout the manuscript.

With regard to “showing gain of function,” it is unclear to us what that experiment would look like and what question it could answer. While a rescue experiment to re-express Nrx1a in adult KOs would have profound implications for treatment, the large size of the Nrxn1a cDNA has impeded our attempts to make a AAV-based viral vector suitable for widespread infection and reliable Nrxn1a expression. In this manuscript we focused on circuit-specific loss-of-function approaches so we could compare them to the constitutive Nrx1a KO mice, as we were interested in modeling the loss-of-function contributions of Nrxn1a to reward processing.

8) Some of the figures should be edited for clarity. For example, in some figures such as 1B, the statistical differences are entirely unclear. Although there is a described overall genotype difference, it is unclear if there was an interaction and at what reward size these differences were statistically important. This lack of clarity can be seen in most of the figures. Further, based on the information presented here, it is not possible to determine interactions or how post-hoc analysis was completed. While the figure legends contain some statistical information, they do not lead the reader to fully understand the important changes. Please also include how any post-hoc comparisons were done.

We have included tables with the outcomes of statistical tests to accompany each figure, for clarity. When developing the protocols tested in this paper, we were interested in exploring whether relative reward discrepancy or probability interacted with mouse genotype to produce a choice deficit in unique reward environments (we hypothesized, for instance, that Nrxn1a mutants would have more severe deficits with increased reward scarcity). Each animal repeated the same task, but with slight variations in reward discrepancy and probability. This repeated design was influenced by our previous paper (Alabi et al., 2019), in which we establish that mice exhibit trait-like rates of performance across multiple sessions in a similar task.

Upon completing our analysis, we found a suspected effect for relative reward discrepancy on performance and reward stay behaviors, but no effect for reward probability. While there was a significant effect for genotype overall, there was no interaction between genotype and the other factors of interest. In other words, the average difference in performance/relative reward-stay between Nrxn1a WT and KO animals did not vary with the discrepancy in reward volume or reward delivery rate. As such, we did not report individual post-hoc tests comparing control and mutant performance in individual reward regimes, as these differences would not be easily interpreted. Additionally, these tests would be conducted using data from individual sessions only, subjecting them to natural session-to-session variability in mouse choice behavior. The repeated measures design was meant to decrease the effects of this variability in mouse behavior and the statistical test that encapsulates that is the ANOVA itself.

One method to better aid clarity is to z-score performance/reward-stay behavior for the population in each reward regime. Condensing z-scores (comparative performance/reward-stay) produces clear differences:

**Author response image 3. sa2fig3:** 

We originally considered showing the data in this manner but decided that too much information was lost in condensing the data down like this.[Editors' note: further revisions were suggested prior to acceptance, as described below.]

Essential revisions:While the revisions have improved the study there are additional revisions needed. Below the remaining concerns are listed including how each should be addressed by tempering the language, re-analysis of data, or the inclusion of one new additional experiment.1) Please address the below comment and comment #2 by performing re-analysis with the collected data and providing additional details. New experiments are not required.Even with the additional control data provided, the fiber photometry data, as presented, is difficult to interpret. The Z-score peaks presented in Figure 7D-F (0.2 Z scores) are significantly lower than peaks seen in the overall representative trace in Figure 7C (where there are peaks of up to ~4 Z-scores). While the authors suggest that the small z-scores are due to genetically defined populations, work from several other groups recording from comparable cell populations have seen much larger task-specific peaks in GCaMP-measured activity (Cui et al., 2013 Nature; Cui et al., 2014 Nat. Prot.; Klaus et al., 2017 Neuron; Calipari et al., 2016 PNAS), demonstrating that this alone cannot explain this. There are several levels of data analysis that are missing from the manuscript that would alleviate concerns regarding this issue:

We are eager to address these points of analysis so that our paper will be clearer to all readers. There are several points that must be first clarified:

a) The Z-score peaks presented in Figure 7D-F are not incongruous with the peaks seen in the overall representative trace in Figure 7C. We note that on our previous submission, Figure 7C does not have a y-axis, so that it is not clear what the absolute magnitude (ΔF/F) of individual peaks are. However, the reviewer correctly points out that there are peaks that are about 4 z-scores greater than the mean/median of the signal. If the mean/median of the signal in Figure 7C were to be less than zero, a peak with a large prominence (that is, with a large amplitude versus surrounding signal) might have a lower absolute magnitude than anticipated. This is the case with the signals that we analyze.

b) When mice are engaged in the task, the average signal (or baseline) is lower than when they are disengaged and running around the cage. Mean signals for animals in engaged and disengaged epochs of the task were calculated. Engaged periods were defined as any timepoint within a window of 5 seconds before an initiation event (to start a trial) and after a nosepoke exit (to end a trial). Epochs outside of this window were defined as periods of disengagement. Note, again, that this phenomenon is seen both in the experiments described in this manuscript, but also in separate experiments in another behavioral context. We note no significant differences in the average signal of control and mutant mice either in the engaged period or the unanalyzed disengaged period. (Analyzed by Two-Way Repeated Measures ANOVA: Engagement, p=0.0003; Genotype, p=0.166; Interaction, p=0.337). We now add this data to Figure7—figure supplement 1.

c) It is important to note that given the lower average signal in engaged periods, the *magnitude* of initiation peaks may not be as high as we expect. Instead, as suggested above, we might expect the *prominence* (height versus surroundings) of initiation peaks to be high. If we look at the fast-peak phase of the large reward initiation trace in Figure 7F, for instance, we can see that while the magnitude of the peak is roughly 0.4 z-scores, this occurs on top of an immediate background of -0.4 z-scores. This means that the dynamic range of initiation-associated peaks is actually 0.8-1 z-scores, on average, rather than the 0.2 z-scores quoted by the reviewer in reference to 7D. This falls within the dynamic range of striatal photometry experiments in the field (and the ones cited by the reviewer).

d) We employed a task with structured behavioral elements that mice had to complete in order to progress with the task. This task structure allowed us to perform analysis of the photometry signal by assessing peri-event striatal dynamics. Note that, for the purposes of this manuscript, we limited our analysis to the pre-initiation time epoch. We have annotated Figure 7C, to highlight a few points. There are 4 initiations shown in Figure 7C. They have been numbered in green, for consistency. A line representing mean value of this trace is also included to represent the baseline of this signal segment, which is, again, less than 0 during this period of engagement with the task.

First, note that the shape of the preinitiation in individual trials roughly approximates the shape of the averaged signals. That is, a downward ramp in activity that precedes a peak in activity, marked with a green timestamp. This holds true even in trial 2, where there is a downward slope followed by a small uptick in activity before the initiation.If we carefully examine the preinitiation slow ramp of each initiation event, we observe that population activity actually drops to either the mean or to a value below the mean value for this trace (which again, is less than 0 to being with). As a result, the subsequent fast peaks associated with these initiations are only slightly greater than the mean value (trials 1,2,4) and in one case actually less than the mean (trial 3). This is one of many reasons that we independently quantified both the ramp and the peak – because both processes can be observed in individual traces and because the two processes together dictate peak magnitudes.As expressed by the reviewer, we note that there are indeed several peaks in Figure 7C of ~4 z-scores. It is important to note however, that those peaks are not associated with initiation events temporally. We performed an analysis on a specific behavioral epoch and that analysis is consistent across figures as demonstrated above. Initiation peak magnitudes are on average ~0.3 z-scores, but the dynamic range of peaks is 0.8-1 z-score. The highest peak values observed in Figure 7C do not occur in phase within our temporal window. These peaks are much more closely associated with nosepoke exits (light blue), occurring either just before or after these events. We further note that, 1) these large peaks exist, even though the data is z-scored and we are recording from a genetically defined cell population and 2) these peaks are not the topic of analysis for this paper because they are not in phase within the temporal window of interest (i.e. they are not associated with initiation events)

e) Finally, we would further emphasize that we purposefully employ a small, but genetically defined, cell population in our photometry experiments. In our previous rebuttal response, we show that in a similar task, in mice in which all DMS D1 neurons are labeled, signal dynamics around initiation are roughly twice as large (1.4-2 scores). The papers cited do not use conditional retrograde labeling of cell populations, instead viral GCamp expression in genetic mouse lines.

a) Changes in Ca^2+^ signaling should be presented in a trial-by-trial basis in addition to the summary seen in Figure 7.b) Trial-specific traces should be presented in a non-Z-scored fashion (see comment 2 for why this is particularly important).

Presented in Author response image 4 are individual GCamp6f traces of trials in a control animal, aligned to initiation, sorted by prior large reward (blue) and small reward (red) trials. The left column are the z-scored data used in the paper and the right column is the raw ΔF/F(%) traces. Note that the trends seen in the averaged z-scored data can be observed in the z-scored data at the individual trial level and also observed in the individual non-z-scored data at the individual trial level. In contrast to the concerns noted below by the reviewer, there is actually a greater dynamic range with z-scoring rather than a flattening of the signal. Finally, comparing the z-scored and non-z-scored data demonstrates the signal drift and shifting baselines normally seen without signal standardization. Averaging at this level produces the same waveform shape but with a larger standard error, as demonstrated in Author response image 5.

**Author response image 4. sa2fig4:** Photometry traces by trial. 10 individual traces selected from trials following large reward (blue) and trials following small reward (red). Panel A and C show z-scored data while B and C show raw ΔF/F(%).

c) Although the data from the 405 control channel is presented, it is presented as an internally controlled Z-score. The 405 traces should be plotted at the same scale as the 470 trace to preclude the signal detected being movement artifacts. There is no mention of scaling in the analysis pipeline and this is critical to using the 405 as a control.

We thank the reviewer for catching a mistake in the labeling of our axis – the 405 data shown here is the actual ΔF/F(%), not a Z-score. We have now made these corrections. In addition, we will clarify our photometry pipeline in the Materials and methods as follows: (1) We did not z-score the 405 signal before incorporating it. The 405 was subtracted from the 470 before z-scoring occurred. We agree with the reviewer that it does not make sense to z-score the 405. (2) The controlFit method for calculating ΔF/F relies on scaling the 405 to the 470, then subtracting the 405 from the 470. We, however, employed another method for calculating ΔF/F. To account for a steady decrease in baseline fluorescence over prolonged sessions, the 405 and 470 were baselined to zero (the last value in the recording was used as an offset to have the signal decay to 0). Following this, the data were fit with a cubic polynomial curve, which was then subtracted from both signals (bleach detrending). Afterward, both signals were standardized by sorting values into a histogram (100 bins) and then selecting the largest bin as the baseline signal. This baseline was subtracted from the raw 470 and 405 and then those values were divided by the baseline (note that the operation below was performed on both 470 and 405). Following this, the control signal was subtracted from the GCamp6f signal [dF/F(a) = (debleach(a)-baseline/baseline]^1^.

d) The authors report the laser power at 50μW, which is very low for both recording channels – especially the 405 – and could in part explain the flat line for the 405 channel. i.e. the light power was not high enough to detect events in that channel.

We note that there are a wide range of light intensities seen in fiber photometry experiments across the literature and at *eLife* itself. *eLife* has published papers in which laser intensity at the tip of the patch cable was approximately 5 μW ^2^ and others where light intensities were about 200 μW ^3^. Our measurement of 50uW falls right into this range of previously published laser light intensities.

e) In the Materials and methods the authors mention that the data is filtered but provide no additional information about the filter. What is the filter, what is the equation, how does this filter alter the timing of the signal relative to the events? This is critical to understand the analysis pipeline.

We use the filtfilt function in Matlab to digitally filter our data. This is a zero-phase digital filter that filters data in both the forward and reverse direction, resulting in zero phase distortion. As a result, it will not change the relationship of the signal to individual behavioral events. This information has been added to the Materials and methods.

2) The biggest issue in the data processing is with how the z-scores were calculated for the peri-event records. Because the signals were done as a change from the entire peri-event trace this makes it so that the median of the trace is 0. For signals with bigger peri-event records this will just move the pre and post response traces down. It will look like the peaks are similar, but the ramp up and down are different. This would then make the data look like the complete opposite of the raw data. The reason this is concerning here is that they data for the largest event shows this pattern, where the baseline is lower, rather than the event. Nearly all of the work on striatal populations has shown that these circuits scale with reward value – with larger peak response around the larger outcome. Thus, these data may actually show the opposite findings when plotted in a more appropriate way.

To address these concerns, we will elaborate on our reasons for z-scoring and establish that this method does not produce alterations in the data.

a) First, we must establish that the signals presented in Figures 7 and 8 were not done as a change in the perievent signal. z-scores were calculated in two-minute windows around each collected sample of the Gcamp signal (one minute in front and one minute behind each point in time). Individual trials only last a few seconds. So when mice are engaged with the task, each data point is z-scored across multiple trials, not 1. This can also be observed in Figure 7C above in which the animal completes 4 trials in a 60 second time period. We performed this z-scoring for the following reasons:

Using this moving window method for z-scoring helped to account for changes in the raw GCamp signal over the course of an individual hour-long session. Along with the de-bleaching methods described above, this method of z-scoring helped to produce a stationary time-series for analysis.By z-scoring data, we could confidently average Gcamp signals over multiple days and multiple mice.Very similar photometry methods have been employed previously in the literature (see^4^).

b) We further note that when the reviewer says “larger peak response around the larger outcome” is established in the literature that:

Our data is consistent with this outcome. Our data demonstrate more dynamic initiation peaks after large rewards. We again emphasize the difference between peak magnitude and peak prominence. Initiation peaks subsequent to a large reward are more dynamic than after a small reward – they have a larger prominence than after small reward outcomes. This larger response occurs against a more silent immediate background, suggesting a greater signal to noise after large reward than after small reward. The relationship between the reward signal and the immediate background signal may be a critical component of encoding reward in this time epoch.While unit recordings show increases in activity with reward, we note again that the object of our analysis is temporally constrained. We are examining initiation events that follow large and small reward outcomes. As such, we are analyzing events that lag reward consumption (this is appropriate as the majority of our behavioral analysis looks at actions that lag outcome by 1 trial). In fact, given the presence of large peaks in our data (such as those seen in Figure 7C and acknowledged by the reviewers in their previous critique), it is possible that larger peak responses may be temporally linked to epochs that occur during and after reward consumption. We might expect larger exit peaks after big rewards for instance.It is unclear whether this comment: *“Nearly all of the work on striatal populations has shown that these circuits scale with reward value”* is referring to activity at task outcome, but if so, there are many examples where this is not the case, both for single unit recordings (see Figure 5 in^5^) and SPN subtype-specific photometry (see Figure 10 in^4^).

c) The main reason that we z-score is to compare different animals across multiple sessions. Not z-scoring, adds more noise to average traces, without changing the relative dynamics of those signals.

**Author response image 5. sa2fig5:** Comparison of raw averaged signals (A) and z-scored signals (B).

The data above clearly demonstrates that the effect of z-scoring is to increase the signal to noise, rather than decrease it. This is mostly a result of allowing us to average the data from multiple mice on multiple days, but there is another effect as well. We have established that there is a moving baseline signal that varies based on the engagement of the animal with the task. Note that when we calculate ΔF/F, we do so using the entire trace. That is, we include both engaged and disengaged periods of the task. z-scoring locally, rather than globally, will thus tend to raise the relative baseline for engaged periods, not lower it, as the reviewer suggests. This is because z-scoring locally will tend to exclude the disengaged periods. The end results are signals with larger magnitude and prominence, as well as less variation, as demonstrated above.

3) Please address the below concern by tempering the language in the manuscript.Although the authors address concerns regarding specificity of the NEX-Cre/Nrx1aKO mouse line and potential mechanistic link between forebrain function in the text, there remains a large disconnect between the widespread loss of neurexin1a expression and the specific effects seen in the DMS->SNr circuitry. As the authors mention, the NEX-Cre has primary recombination sites in the amygdala, hippocampus, and several PFC subregions; each which feed into the DMS-> SNr pathway and each provide a unique aspect to reward-associated behavior. A key premise of the manuscript is that loss of forebrain neurexin1a leads to disruption of the DMS->SNr circuitry during reward-seeking. However, what is missing from the manuscript is how is this circuit disruption occurring. Given the widespread loss of neurexin1a expression, it is difficult to infer a potential mechanism or pathway as a source for DMS activity disruption, beyond likely glutamatergic input.

We have tempered the language and agree that a disconnect remains between the brain region-specific genetic manipulations and the analysis of striatal population signals. However, this is a gap that will not be bridged in any single paper. Our goal here was to demonstrate that given the near brain-wide expression of Nrxn1a, we could in fact localize a reward-processing deficit to Nrxn1a disruption in specific brain structures – a non-trivial feat. We believe the novelty of this itself, coupled with the first analysis of a neuropsychiatric disease model via reinforcement learning paradigms and in vivo imaging of the telencephalic excitatory knockout mice are notable first steps. Furthermore, while we now cite work demonstrating Nrxn1a-associated changes in excitatory transmission within prefrontal- and thalamo-striatal circuits, it would take substantially more work – beyond the scope of this paper – to connect this type of ex vivo acute slice analysis with the altered photometry signals we observe in Nex^Cre^; Nrxn1a^fl/fl^ mice.

4) Please address the below concerns by performing an additional experiment to examine mEPSCs.A main lingering concern is still about the retroAAV cre injection into SNr. The mIPSC recordings in DMS show that one particular change (in inhibitory transmission) does not occur but doesn't rule out other possibilities, particularly in terms of potential changes in excitatory transmission. It is assumed that the mEPSCs in Nex-cre Nrxn1a-cKO mice change in comparison to Nex-cre controls (that is their hypothesis, that corticostriatal deficits occur and underlie the changes in dSPN activity), although they don't show it but reference unpublished data in the reviewer response. A retroAAV cre injection in Nrxn1a cKO-only animals (no Nex-cre) to determine that mEPSCs don't change if the Nex-cre isn't there would be important. This would show that retroAAV cre expression doesn't do anything additional that is not intended at their synapse of interest.

We agree that the complex interconnectedness of the BG is a challenging issue given the manipulation that we had to use to isolate single SPN population signals in the context of our Nex^CRE^;Nrxn1 mutant mice. As such, we have performed the requested experiment to discern whether adult deletion of Nrxn1a from retroAAV2-Cre injections into SNr changes the excitatory synaptic tone onto dorsal striatal dSPNs, the population imaged with photometry in Figures 7 and 8. Into adult Nrxn1l^fl/fl^ mice, we injected rAAV2-3xFlag-Cre (plus DIO-tdTOM reporter virus in dorsal striatum) or rAAV2-EGFP-∆Cre (an enzymatically inactive truncated version of Cre with its own fluorophore) into the SNr, waited the same duration of time as for our behavior/recording analysis (3 weeks) and then recorded mEPSCs in acute slices cut from these mice. We did not observe any significant change in the frequency or amplitude of mEPSCs in neurons with Cre or δ-Cre expression, making it unlikely that adult-mediated alterations in excitation resulting from our retro-AAV2-Cre strategy account for changes in recorded Ca^2+^ signal. These data have been added to Figure 8—figure supplement1. Together with the prior revision’s mIPSC data, we have made a solid case that our technical approach for imaging has not dramatically contributed to the imaging results in Figure 8. We still believe the most straight-forward explanation of our data is that deletion of Nrxn1a from cortical progenitors somehow alters the excitatory synapses of these projections pathways, with one downstream result being the abrogation of value-related signals in a key target, the DMS. We have added a reference to our bioRxiv paper (currently under revision), focused on excitatory synaptic transmission in mPFC- and thalamo-striatal synapses of Nrxn1a ex vivo brain slices. However, we also clarify that given the broad cortical deletion of Nrxn1a with the Nex^Cre^, we are hesitant to try directly linking these results. In fact, as we write in the final paragraph, it will take in vivo recordings of multiple circuit nodes to fully understand how altered striatal signals come about in these mutants – work that is planned for the future.

[Editors' note: further revisions were suggested prior to acceptance, as described below.]

Recommendations:It is suggested that the data presented in the response to reviewers be accessible so that the readers can see the raw data minimally processed. A concern hinges upon data that is included in the response where the z-scored data is introducing changes into the data that is not seen in the raw df/f data. Author response image 5 clearly shows that there are significant differences between A – unprocessed data and B- z-scored data that makes it look like huge decreases are happening that do not seem apparent in the unprocessed data. While this could be fine if the authors are transparent with their analysis pipeline, providing the data would help the reader draw their own conclusions.It is recommended that the authors include this data or make it accessible to the readers in some form (i.e. link, website, etc).

We agree that full transparency is the best solution here. Along those lines, we have posted the entire dataset for this manuscript on Dryad (Alabi, Opeyemi (2020), Neurexin Photometry, Dryad, Dataset, https://doi.org/10.5061/dryad.vhhmgqnrq). Together with the relevant Matlab code posted on the publicly accessible Fuccillo lab Github site (https://github.com/oalabi76/Nrxn_BehaviorAndAnalysis), and the more detailed methods section in the revised manuscript, it should be possible to both recreate our exact analyses, as well as perform any other desired analyses. We have added this information to the modified manuscript.

**References**

1 Holly, E. N. et al. Striatal Low-Threshold Spiking Interneurons Regulate Goal-Directed Learning. Neuron, doi:10.1016/j.neuron.2019.04.016 (2019).

2 Cai, L. X. et al. Distinct signals in medial and lateral VTA dopamine neurons modulate fear extinction at different times. *eLife* 9, doi:10.7554/*eLife*.54936 (2020).

3 Matias, S., Lottem, E., Dugue, G. P. & Mainen, Z. F. Activity patterns of serotonin neurons underlying cognitive flexibility. *eLife* 6, doi:10.7554/*eLife*.20552 (2017).

4 London, T. D. et al. Coordinated Ramping of Dorsal Striatal Pathways preceding Food Approach and Consumption. The Journal of neuroscience : the official journal of the Society for Neuroscience 38, 3547-3558, doi:10.1523/JNEUROSCI.2693-17.2018 (2018).

5 Shin, J. H., Kim, D. & Jung, M. W. Differential coding of reward and movement information in the dorsomedial striatal direct and indirect pathways. Nat Commun 9, 404, doi:10.1038/s41467-017-02817-1 (2018).